# Improving predictability of high ozone episodes through dynamic boundary conditions, emission refresh and chemical data assimilation during the Long Island Sound Tropospheric Ozone Study (LISTOS) field campaign

Siqi Ma[1,2], Daniel Tong[1,3*], Lok Lamsal[4,5], Julian Wang[6*], Xuelei Zhang[3], Youhua Tang[3,6], Rick Saylor[6], Tianfeng Chai[4], Pius Lee[6], Patrick Campbell[3,6], Barry Baker[3,6], Shobha Kondragunta[7], Laura Judd[8], Timothy A. Berkoff[8], Scott J. Janz[4], and Ivanka Stajner[9]

[1]Department of Atmospheric, Oceanic and Earth Sciences, George Mason University, Fairfax, VA 22030 USA
[2]National Research Council, hosted by the National Oceanic and Atmospheric Administration Air Resources Lab, College Park, MD 20740 USA
[3]Center for Spatial Information Science and Systems, George Mason University, Fairfax, VA 22030 USA
[4]Atmospheric Chemistry and Dynamics Laboratory, NASA Goddard Space Flight Center, MD 20771 USA
[5]Universities Space Research Association, Columbia, MD 21046 USA
[6]National Oceanic and Atmospheric Administration (NOAA) Air Resources Laboratory, College Park, MD 22030 USA
[7]NOAA National Environmental Satellite Data and Information Service, College Park, MD 20740 USA
[8]NASA Langley Research Center, Hampton, VA, 23681 USA
[9]NOAA National Weather Service National Centers for Environmental Prediction, College Park, MD 20740 USA

Corresponding authors: Daniel Tong (qtong@gmu.edu); Julian Wang (julian.wang@noaa.gov)

**Abstract.** Although air quality in the United States improved remarkably in the past decades, ground-level ozone ($O_3$) rises often in exceedance of the national ambient air quality standard in nonattainment areas, including the Long Island Sound (LIS) and its surrounding areas. Accurate prediction of high ozone episodes is needed to assist government agencies and the public in mitigating harmful effects of air pollution. In this study, we have developed a suite of potential forecast improvements, including dynamic boundary conditions, rapid emission refresh and chemical data assimilation, in a 3 km resolution Community Multi-scale Air Quality (CMAQ) modeling system. The purpose is to evaluate and assess the effectiveness of these forecasting techniques, individually or in combination, in improving forecast guidance for two major air pollutants: surface $O_3$ and nitrogen dioxide ($NO_2$). Experiments were conducted for a high $O_3$ episode (August 28–29, 2018) during the Long Island Sound Tropospheric Ozone Study (LISTOS) field campaign, which provides abundant observations for evaluating model performance. The results show that these forecast system updates are useful in enhancing the capability of this 3 km forecasting model with varying effectiveness for different pollutants. For $O_3$ prediction, the most significant improvement comes from the dynamic boundary conditions derived from the NOAA operational forecast system, National Air Quality Forecast Capability (NAQFC), which increases the correlation coefficient (R) from 0.81 to 0.93 and reduces the Root Mean Square Error (RMSE) from 14.97 ppbv to 8.22 ppbv, compared to that with the static boundary conditions (BCs). The $NO_2$ from all high-resolution simulations outperforms that from the operational 12 km NAQFC simulation, regardless of the BCs used, highlighting the importance of spatially resolved emission and meteorology inputs for the prediction of short-lived pollutants. The effectiveness of improved initial concentrations through optimal interpolation (OI) is shown to be high in urban areas with high emission density. The influence of OI adjustment, however, is maintained for a longer period in rural areas where emissions and chemical transformation make a smaller contribution to the $O_3$ budget than that in high emission areas. Following the assessment of individual updates, the forecasting system is configured with dynamic boundary conditions, optimal interpolation of

initial concentrations, and emission adjustment, to simulate a high ozone episode during the 2018 LISTOS field campaign. The newly developed forecasting system significantly reduces the bias of surface $NO_2$ prediction. When compared with the NASA Langley GeoCAPE Airborne Simulator (GCAS) vertical column density (VCD), this system is able to reproduce the $NO_2$ VCD with a higher correlation (0.74), lower normalized mean bias (40%) and normalized mean error (61%) than NAQFC (0.57, 45% and 76%, respectively). The 3 km system captures magnitude and timing of surface $O_3$

peaks and valleys better. In comparison with LIDAR $O_3$ profile variability of the vertical $O_3$ is captured better by the new system (correlation coefficient of 0.71) than by NAQFC (correlation coefficient of 0.54). Although the experiments are limited to one pollution episode over the Long Island Sound, this study demonstrates feasible approaches to improve the predictability of high $O_3$ episodes in contemporary urban environments.

**1. Introduction**

Exposure to ambient air pollutants has been associated with detrimental health effects, including cardiovascular diseases and premature deaths (Brunekreef and Holgate, 2002; Kim, 2007; Héroux et al., 2015). Recent decades saw remarkable improvement in the air quality across the United States. From 1990 to 2015, the United States Environmental Protection Agency (US EPA) estimated that the emissions of nitrogen oxides ($NO_x$), a major pollutant that controls regional ozone formation, were reduced from 25.2 to 11.5 million t yr $^{-1}$ (Feng et al., 2020). The downward trends in $NO_x$

emissions have been verified by ground and satellite observations in large cities (Tong et al., 2015) and in the eastern United States (Zhou et al., 2013; Krotkov et al., 2016). Because of the substantial emission reductions, ground-level ozone concentrations decreased ubiquitously across the US (Hogrefe et al., 2011; Simon et al., 2015; He et al., 2020).

    Regardless of the tremendous improvement in air quality, more than one third of the US population still lives in areas exceeding the National Ambient Air Quality Standards (NAAQS) for ozone ($O_3$) and/or fine particulate matter ($PM_{2.5}$)

(US EPA, 2020). Many of these ozone nonattainment areas are located along the northeastern Interstate 95 (I-95, Interstate Highway on the East Coast of the United States) corridor where high density of emissions is produced by transportation and other industrial sources. Surface ozone is formed from photochemical reactions between $NO_x$ and volatile organic compounds (VOCs) (NRC, 1992), and the high emission density of $NO_x$ is a major controlling factor for high ozone events in this region.

As part of the efforts to understand regional $O_3$ pollution, a multi-agency collaborative study of precursor emissions, ground-level $O_3$ formation and transport in the New York City (NYC) metropolitan region and downwind locations, the Long Island Sound Tropospheric Ozone Study (LISTOS), was launched. Extensive measurements were collected between June and September 2018 within the NYC metropolitan area and over the Long Island Sound (LIS). Multiple analyses of the ozone activities during this field campaign have been conducted using numerical models (Baker et al., 2019; Shu et

al., 2019; Berkoff et al., 2019).

    Air quality forecasts are a critical tool used by environmental and public health agencies to mitigate the detrimental effects of air pollution (Eder et al., 2010; Oliveri Conti et al., 2017; Tong and Tang, 2018). Accurate prediction of ambient ozone and its precursors remains challenging due to inherent uncertainties in the model processes (transport, chemistry and removal), as well as in model inputs such as emissions, initial concentrations (ICs) and boundary conditions (BCs).

Prior studies have also revealed that air quality models face additional challenges in predicting surface $O_3$ concentrations

at coastal locations or over complex urban areas, including uncertainties in vertical mixing, deposition processes, spatial-temporal allocation of emissions to the air quality models (Hogrefe et al., 2007; Tong et al., 2006). Therefore, several modeling techniques have been developed to improve the forecasting skills of these air quality models (Liu et al., 2001; Tang et al., 2007). Previous studies (Wu et al., 2008; Sandu et al., 2010) suggested employing data assimilation methods

to adjust the initial conditions of a model to reduce model bias. Optimal interpolation (OI) is a simple data assimilation method used to enhance model prediction (Candiani et al., 2013; Tang et al., 2015; Tang et al., 2017). Considering the modeling sensitivity to BCs, Tang et al. (2009) examined the impact of six different sources of lateral BCs on the CMAQ (Community Multiscale Air Quality) forecast ability and the results showed that using global model predictions for BCs was able to improve the correlation coefficients of surface $O_3$ prediction compared to observations. Evaluations of

different databases and configurations for BCs in short-term and long-term simulations also show that dynamic BCs have a positive impact on numerical predictions (Tang et al., 2007; Makar et al., 2010; Henderson et al., 2014; Khan and Kumar, 2019). However, many of these studies use BCs based on global forecasts of a relatively low resolution (*e.g.,* $1.4° \times 1.4°$ and $2° \times 2.5°$). Therefore, databases with higher resolution, such as satellite observations or regional forecasting products, are introduced to construct boundary conditions that were shown to result in a measurable improvement in model

performance (Borge et al., 2010; Pour-Biazar et al., 2011). Finally, updating emissions from the base year to the specific forecast year has been shown to be an effective approach to reduce the uncertainties of outdated emission inventories to increase forecasting accuracy (Pan et al., 2014; Tong et al., 2015, 2016).

This study examines to what extent can various modeling techniques improve $O_3$ and $NO_2$ predictions over LIS and surrounding areas. As the largest metropolitan area in the United States on the Atlantic Ocean coast, this LIS region

represents one of the most challenging places for air quality modeling. The resolution of the present operational forecasting system, National Air Quality Forecast Capability (NAQFC), operated by the National Oceanic and Atmospheric Administration (NOAA), is at a 12 km horizontal resolution (Davidson et al., 2008). To better resolve fine-scale processes such as sea breeze and recirculation of air pollutants at coastal sites, a high-resolution (3km) air quality forecasting system over the LIS region (LIS3km) has been developed using the latest meteorology and air quality models.

Using observations from ground air quality monitors and the LISTOS field campaign, we evaluate the forecasting skills of the high-resolution air quality forecasting system to predict $O_3$ and $NO_2$ over LIS. Specifically, we use three forecast improvements - dynamic boundary conditions, rapid emission refresh, and chemical data assimilation - to improve the LIS3km system. The effectiveness of each technique to improve forecasting skill is assessed using the observations from the LISTOS and the EPA AirNow network (http://airnowapi.org). Descriptions of the modeling system, forecast

improvements, and observation data are presented in Section 2. Assessments of the CMAQ results with and without different forecast system updates are described in Section 3. The application of the new system to predict a high ozone episode is demonstrated in Section 4A summary of our findings and concluding remarks are provided in Section 5.

## 2. Methodology

**2.1 Study design**


To simulate ozone variability over a complex coastal urban environment, a high-resolution air quality forecasting system has been developed for LIS and surrounding areas. The forecasting system is comprised of state-of-the-science weather, emission, and chemical transport models. The model domain covers eastern Pennsylvania, New Jersey, southern New York, Connecticut and Rhode Island. While this model domain is large enough to capture key physical/chemical

processes within the LIS area, such as sea breeze circulation and photochemistry, the influence of regional transport outside this domain cannot be adequately represented. Therefore, real-time forecasts from the operational NAQFC (Lee et al., 2017), produced by the NOAA National Weather Service, are used to provide dynamic boundary conditions to investigate the effect of this model input on forecasting performance. We also explore the effects of emission adjustment and chemical data assimilation on forecasting performance.


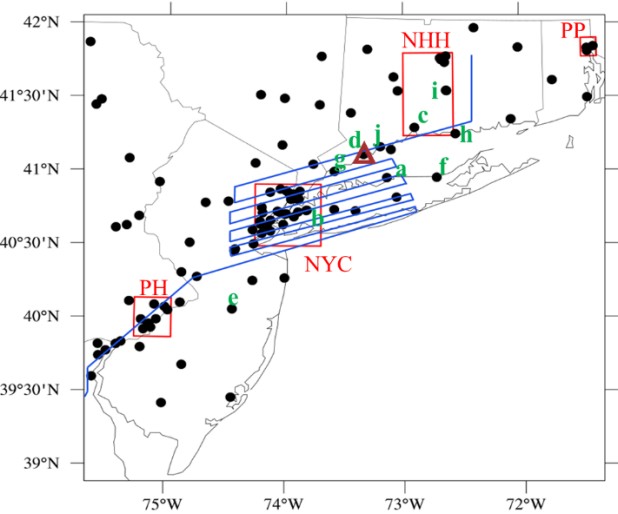

**Figure 1: Study area over the Long Island Sound and surrounding areas. Red boxes depict four subdomains: New York City (NYC), Philadelphia (PH), New Haven-Hartford region (NHH), and Providence-Pawtucket region (PP). Black circles indicate the locations of EPA ground air quality monitors, the brown triangle indicates the TOLNet O$_3$ site located in Westport, CT,**

**and the blue lines present an example flight path conducted by the NASA B200 aircraft on August 28–29, 2018. Letters a–j indicate surface monitoring sites at: a) Flax Pond, b) Queens College, c) New Haven, d) Westport, e) Colliers Mills, f) Riverhead, g) Greenwich, h) Madison-Beach Road, i) Middletown-CVH-Shed and Stratford.**

Five groups of simulations are designed to evaluate the performance and effectiveness of different adjustments of the

CMAQ model (Table 1). The first group (Control run) applies no adjustment, using default profile as LBCs. It serves as the reference case to allow quantifying the effectiveness of each adjustment method. The second experiment, named as BCON, is similar to the Control run, except that dynamic boundary conditions from the NOAA NAQFC with a horizontal resolution of 12 km were applied to replace the default BCs. In the Optimal Interpolation (OI) run, the initial concentrations in CMAQ are adjusted with three observation interpolation methods, including area-average (OI_avg),

inverse distance weighting (idw), and CMAQ concentration gradients (OI_bias) (details of each OI approach provided in Section 2.3(b)). The best performer of these approaches will be used in the subsequent analyses. Next, a group of emission

adjustment experiments are designed to update $NO_x$ emissions using observed changes from satellite and ground sensors (Tong et al., 2016). These emission adjustment factors are applied either uniformly across the domain (EmisAdj_whole), or separately for each subdomain (EmisAdj_sub). In the latter case, the domain was divided into five regions based on

city areas: New York City (NYC), City of Philadelphia (PH), New Haven-Hartford (NHH) and Providence-Pawtucket (PP), and the areas other than these four regions (OTHR) (Fig. 1). Finally, three simulations with the combination of these three techniques were conducted in search of the best performer. All simulations were conducted for a high ozone episode, which lasted 168 hours from 0:00 UTC August 25th to 23:00 UTC August 31st, 2018.

**Table 1. Model adjustment and simulation design for the 3 km forecasting system**

|  | Name | Description |
|---|---|---|
| 1 | Control | Simulation with default profile BCs, no adjustment |
| 2 | BCON | Same as Control, but BCs replaced with NAQFC prediction |
| 3 | OI (3 Cases) | Same as Control, but initial concentrations adjusted by three OI methods (OI_avg, OI_idw and OI_bias) |
| 4 | EmisAdj (2 Cases) | Same as Control, but $NO_x$ emissions adjusted using observed trends from ground and satellite sensors (EmisAdj_avg, EmisAdj_sub) |
| 5 | Combined (3 Cases) | Combination of different techniques. BCON+OI, BCON+OI+EmisAdj_avg, and BCON+OI+EmisAdj_sub |


### 2.2 High-resolution air quality forecasting system (LIS3km)

The high-resolution air quality forecasting system used here is a new research prediction system deployed during the 2018 LISTOS field campaign period which is comprised of three major components: meteorology, emission, and chemical transport models. The Weather Research and Forecasting (WRF) model version 4.0 (Skamarock et al., 2019) is

used to generate hourly meteorological fields to drive emission and air quality modeling. The WRF model was configured with Thompson graupel microphysics scheme, RRTMG long and short-wave radiation scheme, Mellor-Yamada-Janjic PBL scheme, unified Noah land-surface model and Tiedtke cumulus parameterization option. No data assimilation was applied in the WRF simulation. The model is conducted in a single domain with 132×122 grid cells with one grid more on each boundary compared to that of the chemical transport model. There are 41 vertical layers with 20 layers below 1

km and top layer at 50 hPa. The forecast fields of Global Forecast System (GFS) version 4 products with a horizontal resolution of 0.25° × 0.25° (available every 6 h) were employed to drive the WRF model.

The emission input was provided using a hybrid emission modeling system that utilized the Sparse Matrix Operator Kernel Emissions (SMOKE) model (Houyoux et al., 2000) version 4.7 to process anthropogenic emissions, and a suite of emission models to estimate emissions from intermittent and/or meteorology-dependent sources. Anthropogenic

emissions from area and mobile sources were taken from US EPA 2011 NEI version 2 (NEI2011v2). The Motor Vehicle Emissions Simulator (MOVES) was used to generate county-level emission factors for the onroad and offroad sources. SMOKE uses a combination of vehicle activity data, MOVES emission factors, meteorology and other ancillary data (spatial, temporal and speciation information) to generate hourly speciated model-ready emission data. Point sources were

processed in two steps. In the first step, emission inventories of point sources were processed with SMOKE to generate
intermediate input files. Next, these intermediate files were used to drive inline calculation of plume rise to distribute point source emissions vertically in the CMAQ model domain. Two natural sources are included in this forecasting system: biogenic and sea-salt. Biogenic emissions from terrestrial plants were predicted using the inline version of the Biogenic Emission Inventory System (BEIS) (Pierce et al., 1998). The emissions of sea spray aerosols are calculated using an updated version of the Gong (2003) sea-spray emission parameterization (Gantt et al., 2015).

The CMAQ model ingests emissions and meteorology to predict spatial and temporal variations of $O_3$, $NO_2$, and their precursors. In this study, version 5.3.1 of the CMAQ model was configured to include detailed implementation of inline emission processes for biogenic, sea-salt and elevated anthropogenic emissions, horizontal and vertical advection, turbulent diffusion, dry/wet deposition and full gas, aqueous and aerosol chemistry using a revised Carbon Bond 6 gas-phase mechanism and AE6 aerosol mechanism (CB6r3_AE6_AQ) (Byun and Schere, 2006; Luecken et al., 2019). Both
the meteorological and air quality models have a 3 km horizontal resolution over the LIS region and its surrounding areas (Figure 1).

**2.3 Techniques to improve forecasting skills**

We implement and test three forecasting improvement techniques to assess their effectiveness in enhancing the simulation performance of the CMAQ model. Details of each update are described below.

*a) Dynamic lateral boundary conditions*

Regional air quality models such as CMAQ rely on lateral boundary conditions to account for inflow of air pollutants and precursors from out-of-domain sources. These boundary conditions fall into two categories: static and dynamic. Static boundary conditions are time-independent vertical profiles of appropriate species at the boundaries that can be prepared from prescribed profiles, long-term vertical observations, or climatological model simulations (Tong and Mauzerall, 2006;
Tang et al., 2007). Dynamical boundary conditions are provided by a concurrently running global model or another regional model covering a larger domain. In the previous studies of regional modeling, a nested grid approach was often applied to provide dynamic BCs for the study area (*e.g.*, Taghavi et al., 2004). However, the nested model would need higher computational resources and a longer running time. The increasing pool of real-time national and global forecasts provides alternative BCs that can be used to drive a regional forecasting system as demonstrated in this work. Here, we
explore the feasibility of utilizing the products of NOAA NAQFC, which provides real-time national forecasts to prepare dynamic boundary conditions to drive the LIS3km system. The NAQFC is an operational system, operated by the National Weather Services, and the data are provided freely to the public. Hourly forecasts of the NAQFC (Lee et al., 2017) are processed using the BCON tool developed by the US EPA. The description of NAQFC configuration can be found in Lee et al., (2017) and a summary is provided in Table S1.
*b) Optimal Interpolation*

Optimal Interpolation (OI) is a commonly applied data assimilation method (Wang et al., 2013; Chai et al., 2017) that can be used to adjust the initial conditions (ICs) of an air quality model to minimize errors (Adhikary, 2008). This method runs fast and portably, making it very suitable for the forecasting system which needs regular execution. The equation of the OI method is defined as:

$$x^a = x^b + BH^T(HBH^T + O)^{-1}(y - Hx^b) \tag{1}$$

where $x^a$ and $x^b$ are the analyzed and background fields, respectively. $B$ and $O$ are the background and observation error covariance matrix, $H$ is the observational operator and $H^T$ is its matrix transpose, and $y$ is the observation vector.

In the CMAQ model, the restart file, called CGRID, is daily generated during the simulation and acts as ICs for the next day. To constrain the biases in ICs, the concentrations of ozone, $NO_2$ and NO in the restart file were adjusted via the

OI method, which is applied every 24 hours at 0:00 Coordinated Universal Time (UTC). The influence area of OI is controlled by the correlation length scale and the previous study by Chai et al. (2017) chose the range of 84 km for the contiguous US domain. Moreover, this influence length scale also varies from region to region. Over remote regions, the length scale may be longer while it is shorter over polluted areas as the correlation decreases more rapidly. Considering the high emission density and the fine model resolution over the LIS area, we chose a shorter influence length (33km) for

a higher correlation threshold (r >= 0.5) for the LIS which means this OI adjustment was made on each 11×11 grid cell block of the surface layer over the whole domain to obtain the analyzed field $x^a$. Next, as there is no information of vertical background profile in this method, the ratio between $x^a$ and $x^b$ at each surface layer grid point was used to scale the concentrations for all vertical layers within the PBL, following Tang et al. (2015; 2017).

The OI assimilation first allocates ground-based observational data from the EPA AIRNow network into model grid

cells. The Tang et al. (2015) method puts in-situ data directly into the corresponding model grid cells. If there was more than one active site in the same grid cell, the observations are first averaged before being applied to the grid cell (OI_avg hereafter). Grid cells that did not have observations and were not within 5 grids cells from the observations were not adjusted. Therefore, the region of influence is limited, and the adjusted fields may be discrete in spatial distribution. Besides this method, experiments were also performed with two different interpolation methods for preparing the

observational data. The first one was to interpolate the averaged observational grid points to the whole domain using the Inverse Distance Weighting (IDW) interpolation scheme (Shepard, 1968), (the OI_idw method). With this interpolation, the effect of OI will be not limited near the observational sites and most of the grid cells in the domain can be adjusted comparing to the OI_avg. The second method adjusted the initial concentrations by subtracting the bias between the simulation and the averaged observations within the grid point, then smoothing the adjusted concentration field via the

IDW scheme. This method is called OI_bias. Unlike the OI_idw which just applied the spatial interpolation to extend the OI effect, in this method the observation cells are distributed to the whole domain grids based on the spatial patterns provided by the model so that it is able to better reflect the realistic fields.

   c)  *Emission refresh*

The third forecast system update evaluated here is the rapid emission refresh capability that allows timely updates of

outdated NEIs to the forecasting year (Tong et al., 2016). Here we focus on updating $NO_x$ emissions. $NO_x$ are important precursors to tropospheric ozone formation (Spicer, 1983; Chameides et al., 1992), therefore, their emissions can influence atmospheric ozone concentrations. Since $NO_x$ emissions decreased substantially over the last decade (Silvern et al., 2019; Dix et al., 2020) and the anthropogenic emission used in this study are based on the 2011 NEIs, the $NO_x$ emissions need to be projected from 2011 to the forecast year (2018). According to the approach proposed by Tong et al.

(2016), the adjustment factor used for the emission projection is derived from the monthly changing rates of surface- and satellite-observed $NO_x$ ($NO_2$). Temporal trends at the surface are determined from the hourly observed $NO_x$ concentration

during the morning rush hours (06, 07, 08, and 09 local time). These times are optimal for assessing local emission conditions since they are related to the highest $NO_x$ levels typically produced as a result of both commuter traffic peaks and the shallow morning planetary boundary layer (Tong et al. 2015). Satellite-based temporal trends are calculated from the monthly $NO_2$ product retrieved from the Ozone Monitoring Instrument (OMI) aboard the Aura satellite (Lamsal et al., 2020). A weighting function is introduced to combine the surface-based and satellite-based temporal trends to acquire the merged projection adjustment factor (AF) for a specified region:

$$AF = \frac{\Delta S \times N_S \times f_S + \Delta G \times N_G \times f_G}{N_S \times f_S + N_G \times f_G}$$ (2)

where $\Delta S$ and $N_S$ are the temporal trend and the number of satellite data, respectively; and $\Delta G$ and $N_G$ are the temporal trend and the number of surface-based data, respectively. Two weighting factors, $f_S$ and $f_G$ are applied to the satellite and surface data, respectively. Here the value of $f_S$ is set to 1 and $f_G$ to 100 to avoid dominance by either data source (Tong et al., 2015). In this study, two groups of AFs are prepared for the emission projection. One is the average AF over the whole domain (EmisAdj_avg) and the other group includes the AFs for each sub-region in the research area (EmisAdj_sub). The AFs used in both groups are the averages of the monthly AFs from May to September.

## 2.4 Observational data sets

In this study, a suite of observational datasets was used either as inputs for emissions and chemical data assimilation or to evaluate model performance. These datasets include surface $O_3$ and $NO_2$ measurements from the US EPA Air Quality System (AQS) surface network, the $NO_2$ vertical column density (VCD) from the OMI satellite data, $NO_2$ VCD from the GeoCAPE Airborne Simulator (GCAS) on the NASA Langley Research Center B200 aircraft, and the $O_3$ vertical profile from the NASA Langley Mobile Ozone Lidar (LMOL). Detailed information of each data set is provided below.

Surface concentrations of $O_3$ and $NO_2$ are used for emission adjustment and chemical data assimilation, as well as evaluation of model performance. AQS is a routine monitoring network established to collect ambient air pollution data in urban, suburban, and rural areas. AQS monitors determine $O_3$ concentrations according to the Federal Reference Method promulgated in the 2015 revisions to the National Ambient Air Quality Standards (Long et al., 2014) and $NO_x$ concentrations using the chemiluminescence instruments described by McClenny et al. (2002). AQS measures both $O_3$ and $NO_2$ at hourly intervals. Note that $NO_2$ measurements are typically biased high due to interference in the chemiluminescence measurement (Dunlea et al., 2007). As the goal of this study is to improve forecasting performance, a near-real-time version of the AQS data was used, called AirNow. This is a preliminary dataset for the purpose of real-time air quality reporting and forecasting; it is not fully verified and provides fewer measured species. The data used in this study are downloaded from the AirNow data portal maintained by the US EPA.

$NO_2$ VCD measurements were provided by the Ozone Monitoring Instrument (OMI) standard product (version 4), available from the NASA Goddard Earth Sciences Data and Information Services Center (GES DISC). OMI is a nadir-viewing hyperspectral imaging spectrometer that measures solar backscattered radiance and solar irradiance in the ultraviolet and visible regions (270–500 nm) (Levelt et al., 2006). The Aura spacecraft has a local equator-crossing time of 13:45 h in the ascending node. OMI views the Earth along the satellite track with a swath of 3600 km on the surface in order to provide daily global coverage. In the normal global operational mode, the OMI ground pixel at nadir is approximately 13 km × 24 km, with increasing pixel sizes toward the edges of the orbital swaths. Multi-year OMI $NO_2$

data are further aggregated to calculate state-level emission adjustment factors using a mass conservation approach (Tong et al., 2015).

The high-resolution $NO_2$ observations from the GCAS (Kowalewski and Janz, 2014) are used for a direct comparison against model simulations of the $NO_2$ VCD. GCAS is an ultraviolet-visible spectrometer used in air quality field studies to map the spatiotemporal distribution of $NO_2$ and HCHO VCDs at high spatial resolution (Nowlan et al., 2018; Judd et al., 2020). During LISTOS, this instrument flew on 11 flight days collecting between 2-4 gapless raster datasets at spatial resolutions for $NO_2$ as fine as 250 × 250 m. More information about the retrieval can be found in Judd et al. (2020). During LISTOS, $NO_2$ from GCAS was validated using coincident Pandora measurements and had a median percent difference of -1.2% with 95% of the most temporally homogeneous points within ± 25% or 0.1 DU.

Finally, $O_3$ vertical profiles from the NASA LMOL are used to evaluate the CMAQ prediction of $O_3$ profiles during the LISTOS field campaign. LMOL is part of a NASA-sponsored ozone lidar network called the tropospheric ozone lidar network (TOLNet; Sullivan et al., 2017), which is a mobile ground-based ozone lidar platform equipped with a pulsed UV laser and all associated power and lidar control support units (De Young et al., 2017; Gronoff et al., 2019). In this study, we use LMOL lidar observations at the Westport site (41.118° N, 73.337° W). All available field measurement data during this campaign were obtained from the LISTOS Data Archive (https://www-air.larc.nasa.gov/missions/listos/index.html).

**3. Evaluation on the effectiveness of simulation improvements**

**3.1 Effects of boundary conditions**

In this section, we examine the effects of using the dynamic boundary conditions on $O_3$ and $NO_2$ predictions. As a reference, we also compare these simulations to the NAQFC results, extracted for the same region, during the August 29 high ozone event. Figure 2 shows the $O_3$ and $NO_2$ 24-hour average concentrations simulated by Control (static BCs), BCON (dynamic BCs) and the NOAA NAQFC over the LIS region. Comparing to the underestimated $O_3$ concentrations simulated by Control run, the concentration level using dynamic boundary conditions increases considerably and is closer to the observations. High $O_3$ concentrations appear over near-coast areas, but are lower in the northwest of the domain. This spatial pattern illustrates the ozone river in a northeastward direction along the I-95 corridor, extending from Philadelphia to NYC, and then to Connecticut where the worst air quality is often observed. Although it overestimates surface $O_3$ in Philadelphia and central New Jersey, the BCON simulation can reproduce $O_3$ hourly variations during this episode well in comparison with the observed data (see the time series in Figure 2d). Note the peak $O_3$ simulated in the Control run is nearly the same on all days during the simulation period. The comparisons between the peak $O_3$ with the default profile and dynamic LBC case indicates relatively large regional contributions on these days. Compared to the Control run, the BCON run performed better not only in bias, but also with higher correlation coefficients between prediction and observations (Table S2), especially during the August 26–27 high $O_3$ days. As the profile BCs are static and lack spatial-temporal variations, the Control run mainly reflects the local contributions of emissions, transport and chemical processes within the domain (Tang et al., 2007). The underprediction suggests that these processes are insufficient to produce the observed $O_3$ levels, and that the transport of air pollutants from upwind is important to predict

the high O₃ episodes. It highlights the significant influence of dynamic BCs on the simulations over this region during high pollution events. In comparison, the influence of BCs is less important during the cold season. There is a smaller difference between upwind concentrations and the background concentrations used in the default BCs, compared to that during a hot season when the upwind photochemical production is more active, resulting in better agreement between the prediction and observations (Fig. S2a, d). Note that other studies have shown the influence of BCs can become more prominent during the cold season when "local" pollution production is slow (e.g., Fiore et al., 2009). The magnitude of the actual influence is determined by several factors, such as emission density, photochemical production and sink, and spatial distribution and gradients of the concerned species.

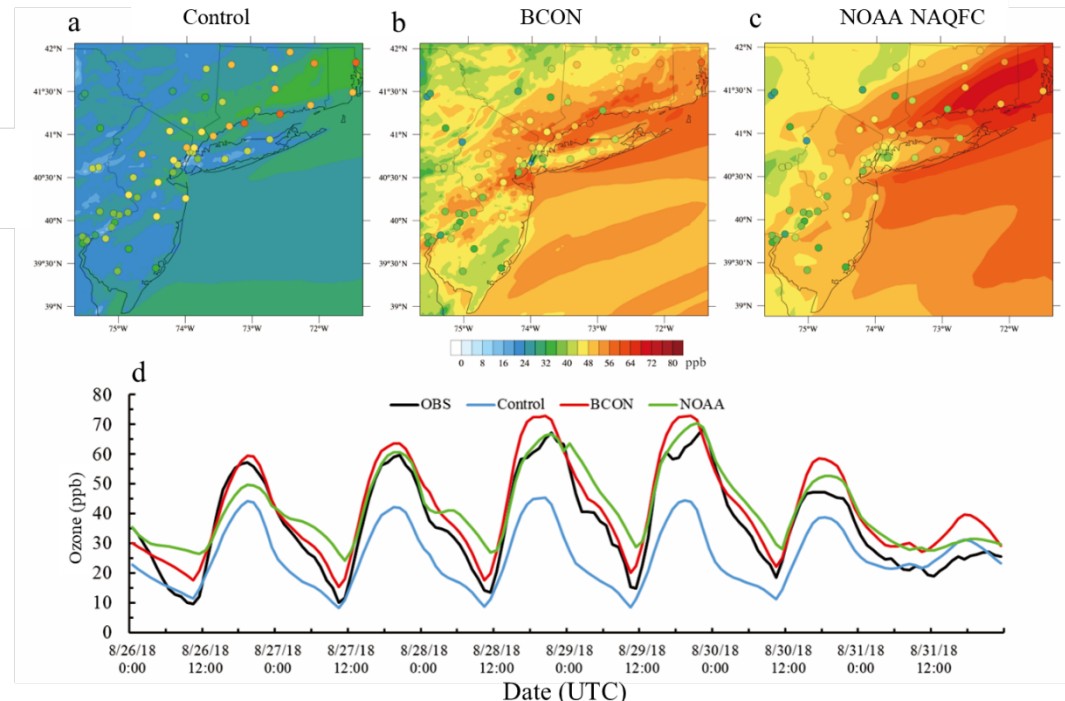

**Figure 2: Predicted O₃ concentrations from (a) Control, (b) BCON and (c) NOAA NAQFC simulations on August 29, 2018, and (d) comparison of domain-averaged hourly O₃ concentrations with EPA AirNow measurements during this episode. Colored circles at the top panels depict the observed concentrations from ground measurements.**

The performance of the high-resolution simulation is next compared to that by the NAQFC. The NAQFC simulation, which has been used to provide national numerical guidance for O₃ and PM₂.₅ (Lee et al., 2017), is run at a coarser resolution (12 km), using a different CMAQ version (a revised CMAQ5.0), driven by different emission and meteorology datasets. Regardless of these differences, the NAQFC and BCON runs predict similar surface O₃ distribution patterns. Compared to that in the NAQFC prediction, the O₃ prediction from the 3 km BCON run demonstrates more detailed spatial distribution. For instance, the O₃ concentration over the Long Island Sound is lower than its surrounding areas during this episode, which is better resolved by the 3 km simulation than the 12 km NAQFC (Figure 2a-c). The O₃ distribution along the coastal area, such as the coasts of Connecticut and Rhode Island, also agrees better with the observations than the 12 km NAQFC prediction. This suggests that the high-resolution simulation can better reproduce the pollutant variability over this coastal urban area during this episode. In addition, the BCON run performs better over southern New Jersey, and northeast of the LIS domain, with considerably reduced biases in the LIS downwind areas as

well. As to the diurnal variations, the BCON run overestimates the peak $O_3$ concentrations on August 28 and 29, while the NAQFC run performs well and is closer to the measurements (Fig. 2d). The use of coarser resolution NAQFC predictions as BCs substantially improves the capability of the 3 km forecasting system to reproduce the $O_3$ variability. Compared to the Control run, the correlation coefficient between BCON and observed $O_3$ concentrations increases from 0.81 to 0.93 and the relative mean square error (RMSE) decreases from 14.97 ppbv to 8.22 ppbv with a reduction of 45%, resulting in a comparable performance with the NOAA NAQFC predictions with correlation of 0.91(Table S2).

The spatial patterns of predicted $NO_2$ concentrations from the Control, BCON, and NAQFC runs are quite similar with high values over the NYC area (Fig. 3). The simulated $NO_2$ concentrations by the 3 km forecasting system, either with static or with dynamic BCs, agree better with the observations than those from the 12 km NAQFC simulation, highlighting the importance of using high resolution inputs to better represent the emission sources in the model. The correlation coefficient and RMSEs are 0.69 and 4.12 ppb for the Control run, 0.71  and 3.82 ppb for the BCON run, and 0.67  and 4.98 ppb for the NAQFC run, respectively (Table S2). In addition, the improvement of simulated $NO_2$ using dynamic BCs was much smaller compared to that of $O_3$. This is because the lifetime of $NO_2$ is relatively short (1–7 h in summertime, Lu et al., 2015), and its budget in urban areas is mainly influenced by local emissions and chemistry, and less by regional transport, indicating the effectiveness of dynamic BCs depends not only on the downwind/upwind gradients, but also on the lifetime of concerned species.

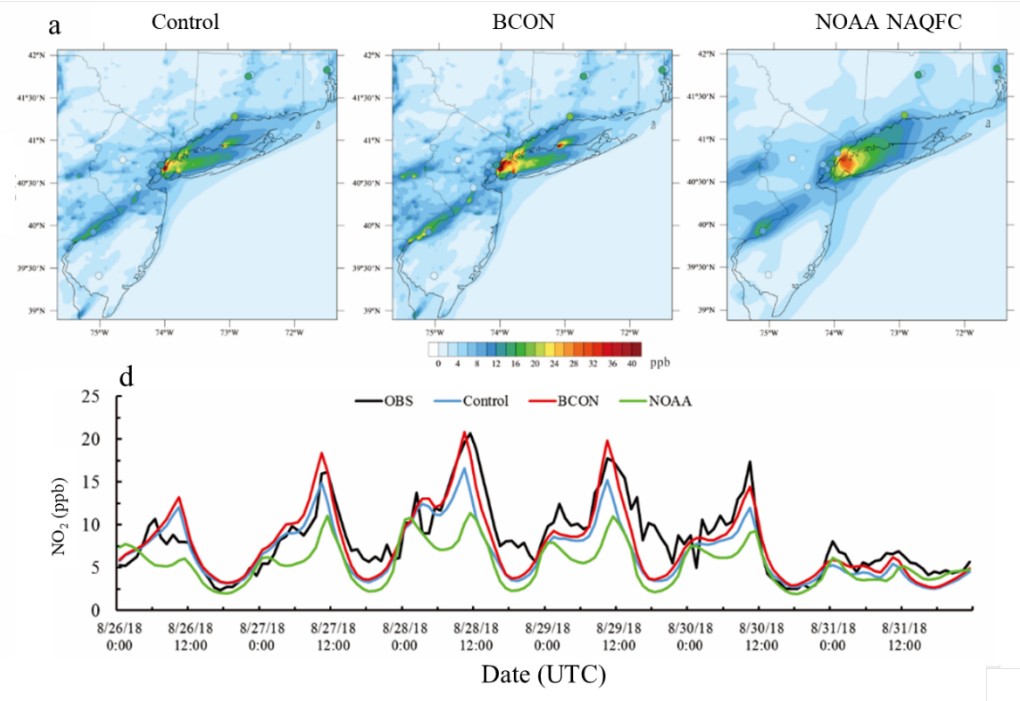

**Figure 3: Predicted $NO_2$ concentrations from (a) Control, (b) BCON and (c) NOAA NAQFC simulations on August 29, 2018, and (d) comparison of domain-averaged hourly $NO_2$ concentrations to EPA AirNow measurements during the episode. Colored circles at the top panels depict the observed concentrations from ground measurements.**

**3.2 Effects of initial condition adjustment**

Initial concentrations are an important input to air quality forecasting. Adjusting initial concentrations through chemical data assimilation has been shown to significantly improve air quality forecasting (Tang et al., 2015; Chai et al., 2017) although the impacts wane with increasing forecast length. Here we compare the results using various OI methods with the simulations without any BC adjustment (same as the Control run) and study the effects of adjusting initial conditions on $O_3$ and $NO_2$ prediction.  Figure 4 illustrates the initial concentrations of surface $O_3$ adjusted by OI_avg,

OI_idw and OI_bias, respectively. In the initial concentrations, the areas influenced by OI_avg are primarily limited to the ground-based sites and the regions within five model grid cells in each direction of the observations compared to the Control run (Fig. 4a, b). The rest of the domain is not affected by the adjustment, resulting in significant differences between adjusted and unadjusted areas. The $O_3$ fields adjusted by OI_idw (Fig. 4c) and OI_bias (Fig. 4d) show similar horizontal distributions, but the concentration level of OI_bias is relatively higher over NYC and northern New Jersey.

Furthermore, in contrast to the localized changes by OI_avg, those of OI_idw and OI_bias show smoother changes over larger parts of the domain.

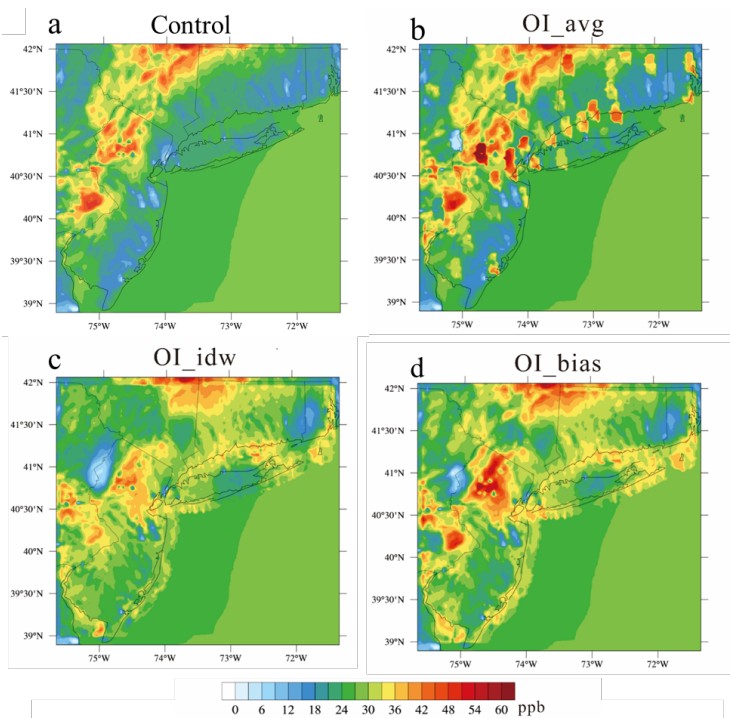

**Figure 4: The concentrations of surface $O_3$ in initial conditions file at 00 UTC August 26, 2018 adjusted by OI_avg, OI_idw and OI_bias.**

       Next, the initial concentrations files after adjustment are used to feed CMAQ simulations. The $O_3$ prediction by the Control run and three OI runs at 00:00 UTC on August 26, 2018 (the first hour after OI adjusting) are depicted in Figure

5. The adjusted $O_3$ fields show different patterns compared to that in the Control run with no IC adjustment. The predicted $O_3$ field with the OI_avg method shows a distribution with localized high value areas near the observational sites. As for the other two OI methods, the distribution using OI_bias has similar patterns with that of OI_idw while the concentrations

over the high $O_3$ area are further elevated. Biases between observed and predicted concentrations are reduced in most areas. The statistical metrics calculated from hourly simulated and observed data from August 26 to 31, 2018 are reported

in Table 2. The RMSEs for $O_3$ are reduced from 14.97 ppbv in the Control run to 13.72 ppbv in the OI_bias run, to 13.79 ppbv in the OI_idw run, and to 14.30 ppbv in the OI_avg run. The correlation for $O_3$ also slightly increases from the Control run to the OI runs (Table 2). In comparison, $NO_2$ prediction is less influenced by this adjustment, with insignificant changes in the model performance (Table 3). In addition, the effects of this adjustment on the modeling results decrease with the simulation time and display no discernible difference from the Control run after 12 hours (Fig.

S1). Generally, the improvement of the simulated results due to OI data assimilation over the study domain is smaller than that from the dynamic BCs. Among the three OI methods, the simulation with OI_bias shows the best performance, so this method is chosen for subsequent analyses in which multiple techniques are combined to improve forecasting skills.

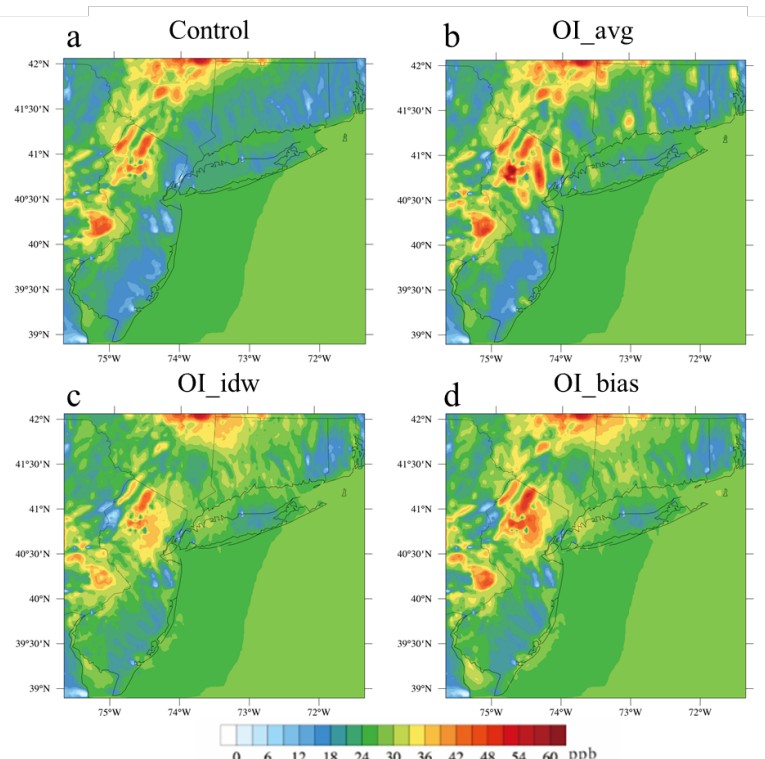

**Figure 5: Spatial distributions of predicted surface $O_3$ concentrations using three Optimal Interpolation (OI) approaches (OI_avg, OI_idw, and OI_bias) at 00 UTC on August 26, 2018.**

**Table 2: Regional mean statistical metrics between hourly observed and simulated $O_3$ from August 26 to 31, 2018 over the Long Island Sound region**

| Stats\Runs | Control | OI_avg | OI_idw | OI_bias |
|---|---|---|---|---|
| CORR | 0.81 | 0.84 | 0.85 | 0.85 |
| RMSE | 14.97 | 14.30 | 13.79 | 13.72 |
| NMB | -30% | -29% | -27% | -27% |

| | | | | |
|---|---|---|---|---|
| NME | 34% | 33% | 31% | 31% |

CORR: correlation coefficient, RMSE: relative mean square error, NMB: normalized mean bias, NME: normalized mean error.

**Table 3: Same with Table 2 but for NO$_2$**

| Stats\Runs | Control | OI_avg | OI_idw | OI_bias |
|---|---|---|---|---|
| CORR | 0.69 | 0.69 | 0.69 | 0.70 |
| RMSE | 4.12 | 4.11 | 4.08 | 4.08 |
| NMB | -17% | -17% | -15% | -17% |
| NME | 35% | 35% | 35% | 34% |

The ICs for each day were adjusted by OI using real time observations, it is interesting to note that the duration of OI influence on O$_3$ simulation varies from place to place. Figure 6 shows the time series of the averaged differences in predicted hourly O$_3$ concentrations between the Control run and each of the three OI runs from August 26 to 31, 2018 in three urban areas (NYC, Philadelphia, New Haven – Hartford) and other (OTHR) areas. The differences illustrate the effect of adjusting initial concentrations on O$_3$ prediction. In large metropolitan areas, OI adjustments result in spikes that indicate larger model errors at the time of OI adjustment, with the mean errors up to 14 ppbv in surface hourly O$_3$ concentrations over NYC and 16 ppbv over Philadelphia, respectively. In comparison, the spikes in non-urban areas are much smaller, reflecting the fact that there are smaller biases between observations and predictions (Fig. 6). The New Haven–Hartford region sees a smaller change of O$_3$ concentration compared to between that in large cities. The OI effects in large cities remain for a shorter time than in non-urban area or smaller cities. For example, the differences between OI runs and the Control run decrease to ~0 ppb in four to eight hours in two metropolitan areas, NYC and Philadelphia (Fig. 6a, b). Meanwhile, in the New Haven – Hartford region (Fig. 6c), Providence-Pawtucket region (not shown) and the non-urban areas (Fig. 6d), the differences could last 12 to 16 hours. The different durations indicate the influence time of OI adjusted ICs, not necessarily the improvement in model skill, which is determined by both initial concentrations and other processes (chemical production and transport, etc.). The improvement using OI adjustment is comparable over different subdomains (Table S3). This difference reflects the dependence of O$_3$ level on the initial concentrations in the air quality model. In general, the influence of OI adjustment lingers for a longer period in an area with low emission density where emissions and chemical reactions make a smaller contribution to the O$_3$ budget than that in the area with high emission density.

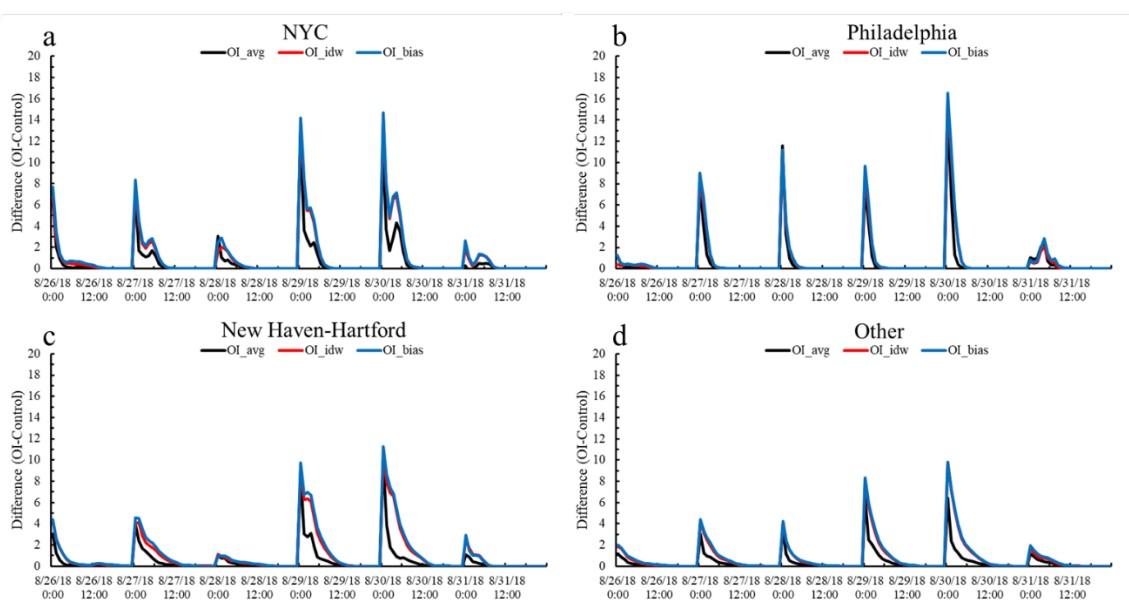

**Figure 6: Effects of OI adjusted initial concentrations on hourly surface O₃ in three metropolitan areas (New York, Philadelphia, and New Haven-Hartford) and the rest of domain using three Optimal Interpolation (OI) approaches (OI_avg, OI_idw, and OI_bias).**

### 3.3 Effects of NOₓ emission adjustment

One of the major challenges in air quality forecasting is the time lag in updating the emission inputs generated for a specified base year which is typically different than the year for which the simulation is desired (Tong et al., 2012). Here we test the effects of implementing a new emission update technique, the rapid emission refresh, on forecasting performance. In this study, the NEI2011v2 data are used to represent anthropogenic emissions, while the target forecasting year is 2018. Both the AQS ground monitors and the OMI sensor observed considerable decreases in $NO_x$ during summertime (May-September) from 2011 to 2018 (Fig. 7). The largest reduction in ground concentrations appears in the west of NYC. The OMI $NO_2$ observations show an increase primarily over Connecticut and Rhode Island, the region downwind of the Long Island Sound (Fig. 7b). The average AF for the whole domain is -18.6%. The AFs for each subdomain are -31.9% for NYC, -12.7% for Philadelphia, -9.4% for the New Haven – Hartford region, -28.2% for the Providence-Pawtucket region, and -16.5% for other regions, respectively. In general, the $NO_x$ variations in this study are similar to that between 2005 and 2012 (Tong et al., 2015), indicating that the $NO_x$ emissions continued decreasing during the past 14 years. This trend highlights the importance of updating the emissions to the model year, in order to reduce the bias in the emission inputs for model simulations, especially for time-sensitive applications such as air quality forecasting.

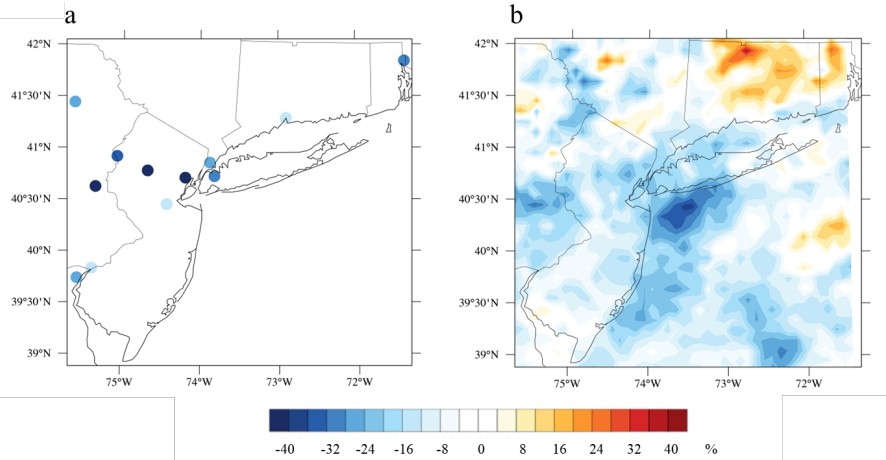

**Figure 7: NO$_x$ differences observed by (a) AQS and (b) OMI from summer 2011 to summer 2018 over the LIS model domain.**

The results in Table 4 and 5 show that the performance for O$_3$ and NO$_2$ prediction is very similar between two simulations using the two emission adjustment methods (a uniform average adjustment factor over the entire domain, and spatially varied factors for each subdomain defined in Figure 1). The correlations in each sub-domains are the same and the average for both simulations is 0.81 for O$_3$ and 0.69 for NO$_2$, respectively. The biases and errors are also at the same level from the two simulations. Compared to the O$_3$ in the Control run, RMSE changes slightly from 14.97 ppbv to 14.71 ppbv (EmisAdj_avg) and 14.55 ppbv (EmisAdj_sub), while the correlation remains the same. The largest differences appeared in NYC with RMSE of 15.54 (EmisAdj_avg) and 14.93 ppb (EmisAdj_sub). This demonstrates that emission adjustment alone results in limited improvement of O$_3$ prediction, due in part to the fact that the O$_3$ production in this region is NO$_x$ saturated (VOC limited) in urban areas where most AQS monitors are deployed, so the O$_3$ level is less sensitive to the change in NO$_x$ emissions. Similarly, the retrievals of satellite observations are also more sensitive to urban plumes. In addition, regional transport of air pollution results in dispersion of emitted NO$_x$ and its byproducts/reservoirs. The observations from satellite or ground monitors, based on which the emissions were adjusted, may not accurately capture the temporal evolution of the emission sources. A large geographical range may better reflect the overall changes of NO$_x$ emissions in the LIS region. Previous studies either use a coarse model resolution (e.g., 1 degree in Lamsal et al., 2011, or state-level adjustment in Tong et al., 2016). As a result, the simulated concentrations using different methods were very close and the limited difference can also get averaged out when calculating the averaged statistical metrics. The effect of the emission adjustment method in this study is not as large as BCON or OI adjustments, which directly influence O$_3$ concentrations. A recent study by Jin et al. (2020) showed that the decrease in NO$_x$ emissions has shifted the NO$_x$-saturated to NO$_x$-sensitive regime transition zone closer to urban centers, approximately 40 to 60 km from the center (the highest emission point) of New York City. Therefore, it is expected that the effectiveness of emission adjustment will increase over time in this region. For surface NO$_2$, the emission adjustment showed a more significant impact on simulated concentrations. Note that the emission adjustment was only implemented in the LIS system, not in NAQFC, which still uses the 2014 NEI for anthropogenic emissions. Without the emission adjustment, the changes in NO$_x$ emissions between the inventory and forecast years are not accounted for. On the high O$_3$ days, NAQFC over-predicted surface O$_3$ during the study period (Fig. 2c). The NAQFC LBCs are likely associated with a possible over-prediction of the regional transport, which can be partially responsible for the BCON LIS simulation overpredicting O$_3$

during high $O_3$ days (Fig 2d). Considering the similarities of these two emission adjustment methods, they will be both tested in the subsequent multi-adjustment simulations.


**Table 4: Statistical metrics of $O_3$ prediction performance after $NO_x$ emission adjustment in different sub-regions from August 26 to 31, 2018**

| Domains/Stats | EmisAdj_avg | | | | EmisAdj_sub | | | |
|---|---|---|---|---|---|---|---|---|
| | CORR | RMSE | NMB | NME | CORR | RMSE | NMB | NME |
| NYC | 0.78 | 15.54 | -34% | 36% | 0.78 | 14.93 | -32% | 35% |
| PH | 0.78 | 15.29 | -30% | 35% | 0.78 | 15.38 | -31% | 35% |
| NHH | 0.85 | 13.24 | -25% | 31% | 0.85 | 13.24 | -25% | 31% |
| PP | 0.81 | 17.26 | -31% | 35% | 0.81 | 17.06 | -30% | 34% |
| OTHR | 0.84 | 12.24 | -24% | 29% | 0.84 | 12.17 | -24% | 29% |
| Average | 0.81 | 14.71 | -29% | 33% | 0.81 | 14.55 | -28% | 33% |

**Table 5: Same as Table 4 but for $NO_2$**

| Domains/Stats | EmisAdj_avg | | | | EmisAdj_sub | | | |
|---|---|---|---|---|---|---|---|---|
| | CORR | RMSE | NMB | NME | CORR | RMSE | NMB | NME |
| NYC | 0.82 | 4.23 | -22% | 27% | 0.82 | 4.77 | -29% | 31% |
| PH | 0.79 | 5.69 | -36% | 41% | 0.79 | 5.53 | -33% | 40% |
| NHH | 0.49 | 7.69 | -44% | 49% | 0.49 | 7.53 | -41% | 48% |
| PP | 0.67 | 2.92 | -18% | 35% | 0.67 | 2.95 | -21% | 36% |
| OTHR | 0.69 | 2.56 | -33% | 39% | 0.69 | 2.54 | -32% | 39% |
| Average | 0.69 | 4.62 | -31% | 38% | 0.69 | 4.67 | -31% | 39% |

## 3.4 Effectiveness of combined adjustment methods

After assessing the effects of individual updates, we test how these updates can be combined to optimize forecasting performance. In the preceding sections, three groups of adjustment approaches have been included and evaluated. For
each group, the best performing method has been identified, including the dynamic BCs, ICs with OI-bias, and rapid emission refresh (including EmisAdj_avg and EmisAdj_sub). With these selected updates, we design and conduct two multi-adjustment simulations, the first one used both the dynamic BCs and the OI-bias adjusted initial concentration files (BO for short) and the other one employed the $NO_x$ emission adjustment together with the combination of BCON and OI-bias (BOE hereafter). Results of these combined adjustments are compared against the Control, BCON run and the
NAQFC prediction.

First, we compare two BOE simulations, one with the EmisAdj_avg emission adjustment and the other with EmisAdj_sub. The statistical metrics of BOE with EmisAdj_avg and BOE with EmisAdj_sub (Table S4, S5) are quite similar in each sub region and also have the same correlations. On average, the RMSEs of the combined BOE setup using the EmisAdj_avg method are slightly smaller than that using the EmisAdj_sub method during the study period (Table S4,
S5), which is different from that when a single adjustment method was applied (see Tables 4 and 5). Therefore, in the

subsequent evaluation we take BOE (EmisAdj_avg) to compare against surface and other observations. Figure 8 compares the predicted hourly $O_3$ and $NO_2$ concentrations against in-situ observations from August 26 to 31, 2018 in five subdomains and the overall domain with Taylor diagrams (Taylor, 2001). In the Taylor diagram, the relative skill of each forecasting system to reproduce the $O_3$ and $NO_2$ variability is represented using three statistical metrics: correlation (R)

with values on arc of the right angled sector, normalized standard deviation (SD) with values on y-axis, and centered root-mean-square difference (RMSD) with values on x-axis. The normalized SD is shown as the dashed line concentric circles while RMSD is shown as line concentric circles with the observation point acting as center (OBS on the x-axis). Their values higher (lower) than 1 indicate biased high (low) of the simulations. In general, the forecasting skill is measured by the distance to the OBS point on these diagrams, the shorter the better. The default (Control) run yielded a correlation

coefficient of approximately 0.8 (0.77–0.84) in each subdomain while those with adjustments show stronger correlations with R all above 0.9. Furthermore, the performance in the OTHER areas is better than that in the five subdomains with the R value up to 0.97 and SD close to 1 (Fig. 8e). Taylor diagrams also reveal that these adjustments are even more effective over the low emission areas. The three adjusted runs, namely BCON (#2), BO (#3) and BOE (#4) in the diagrams, have well reproduced surface $O_3$ concentrations over the NYC region. The simulations with BOE usually demonstrate a

relatively lower $O_3$ concentration level than that with the BCON run or the combined BCON and OI run. This means in the overestimated areas (such as NYC, Fig. 8a), the simulations with emission adjustment show better performance than that without emission adjustment. In addition, these three simulations have similar biases and errors with NMB ranging from 4% to 22% and NME from 15 to 22% (Fig. 9a, 9c). These results illustrate the importance of combining complementary modeling system updates to reduce model uncertainties in a comprehensive way. A single update, such

as emission adjustment, may result in a better emission input closer to the "true" level, but its effect can be offset by systematic biases caused by other inputs. Concurrent improvements of boundary conditions and initial concentrations allow a more realistic initial state and boundary conditions to demonstrate the effectiveness of the emission adjustment in improving $O_3$ forecasting (Fig. 9).

    The Taylor diagrams show that the performance of variability of $NO_2$ predictions is generally worse than that of

variability of $O_3$ predictions. Overlaid on the same diagrams, the points that represent $NO_2$ performance are all further away from the OBS point compared to that representing $O_3$ from the same simulations (Fig. 8). This is not surprising as $O_3$ has been one of the focal points in air quality modeling in the past decades, while $NO_2$ has not been scrutinized with the same intensity. All of the high-resolution simulations, including the Control run with unrealistic boundary conditions, perform better for $NO_2$ prediction than the NAQFC run (Fig. 9), highlighting the benefit of using a high-resolution

modeling system for predicting short-lived chemical species such as $NO_2$. The NAQFC generally underestimates $NO_2$ concentrations in all subdomains. Its bias is the smallest in the NYC subdomain and the largest in the downwind New Haven-Hartford region. The correlation coefficient is between 0.8 and 0.9 in NYC, but lower than 0.6 in the New Haven-Hartford region (Fig. 8). Similarly, the NMB are within 10% in NYC but can be as large as -65% in the New Haven-Hartford region. Such a contrast suggests either an underestimate of emission sources in Connecticut, or an unrealistically

short lifetime of $NO_x$ due to flawed model chemistry, or a combination of both.

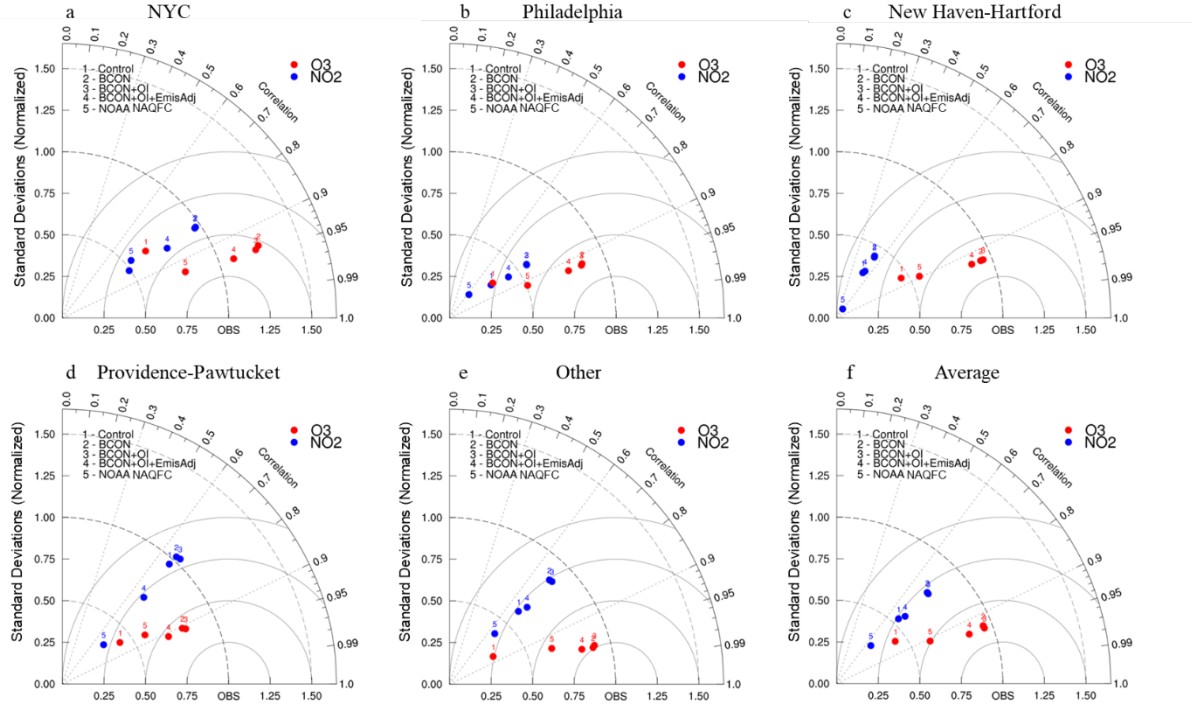

**Figure 8: Model performance in Taylor diagrams of hourly O₃ (in red color) and NO₂ (in blue color) simulated by five runs, including the Control run, dynamic boundary conditions (BCON), boundary conditions with optimal interpolation (BCON+OI), and an all adjustment run including emission adjustment (BOE), and the operational NOAA national air quality forecast capability (NAQFC) run during the episode over five subdomains and the overall domain (Average). The comparison time is from August 26 to 31, 2018.**

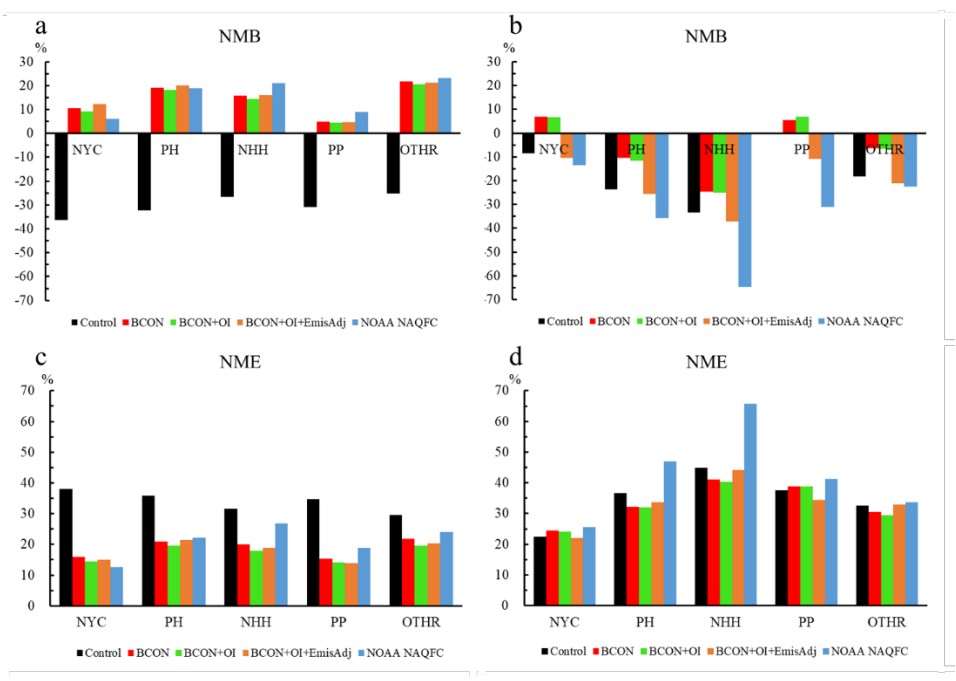

**Figure 9: Comparisons of model performance for surface O$_3$ (top row) and NO$_2$ (bottom row) concentrations from five CMAQ simulations against measurements from the Air Quality System monitors. These simulations include the Control run, dynamic boundary conditions (BCON), boundary conditions with optimal interpolation (BCON+OI), and an all adjustment run including emission adjustment (BCOI+OI+EmisAdj), and the operational NOAA national air quality forecast capability (NAQFC) run during the episode over five subdomains. Two performance metrics are used here: normalized mean bias (NMB) and normalized mean error (NME). The comparison time is from August 26 to 31, 2018.**

## 4. High O$_3$ episode simulations during the LISTOS field campaign

In this section, the newly developed high-resolution system, equipped with all forecast improvements (dynamic Boundary conditions, Optimal interpolation, and Emission adjustment, or BOE), is used to simulate a high O$_3$ episode over the Long Island Sound region. During the high O$_3$ pollution days (August 28-29, 2018) in this episode, surface O$_3$ concentrations exceeded the National Ambient Air Quality Standard (NAAQS) (daily maximum 8-hour average of 70 ppbv) at several monitoring locations, including one site (Colliers Mills) in New Jersey, one site (Riverhead) in New York, and five sites (Greenwich, Madison-Beach Road, Middletown-CVH-Shed, Stratford, and Westport) in Connecticut. While merely exceeding the threshold values by a few ppbv at most sites, the O$_3$ concentrations reached 84 ppbv at the Westport site, and 87 ppbv at the Stratford site. Considering the significant emission reduction and air quality improvements in the eastern United States (He et al., 2020; Qu et al., 2019), this episode, which occurred during a well-designed field campaign, offers a rare opportunity to assess how well a state-of-the-science air quality model can predict a high O$_3$ pollution event that is now less frequent than in the past decades.

### 4.1 NO$_2$ prediction

CMAQ predictions of NO$_2$ surface concentrations and vertical column density are compared against ground and aircraft observations. NO$_2$ is not only a key precursor to tropospheric ozone, but also a proxy for traffic-related air pollution in many epidemiological studies (e.g., Jerrett et al., 2007). Within the LISTOS CMAQ domain, there are four active ground monitors with valid NO$_2$ readings during the study period. Hourly variations from AQS monitors, the BOE 3 km prediction, and the operational NAQFC prediction are illustrated in Fig. 10. Among these sites, the lowest NO$_2$ concentrations were observed at the Flax Pond site in the middle of Long Island, away from the major emission sources. Both BOE and NAQFC are able to reproduce the magnitude and diurnal variations of surface NO$_2$ concentrations at this site. The NO$_2$ concentration at the Queens College site, also located in the Long Island Sound and downtown NYC, is significantly higher than at the Flax Pond site, due to its close proximity to major sources such as the tunnels, harbors and highways. At this site, the BOE 3 km prediction is considerably better than that from the NAQFC prediction. Similarly, the BOE prediction outperforms the NAQFC at the New Haven site in Connecticut, where the surface NO$_2$ concentration reaches 40 ppbv on August 28 and 55 ppbv on August 29, 2018. The NAQFC predicted concentration is constantly below 10 ppbv, severely underestimating the observations. In comparison, the BOE predicted concentrations are much closer to the observations, although still underpredicting the latter. Finally, both models missed the first, primary peak on both days at the Westport, CT site, which is strongly influenced by the NY City plume and sea breeze circulation.

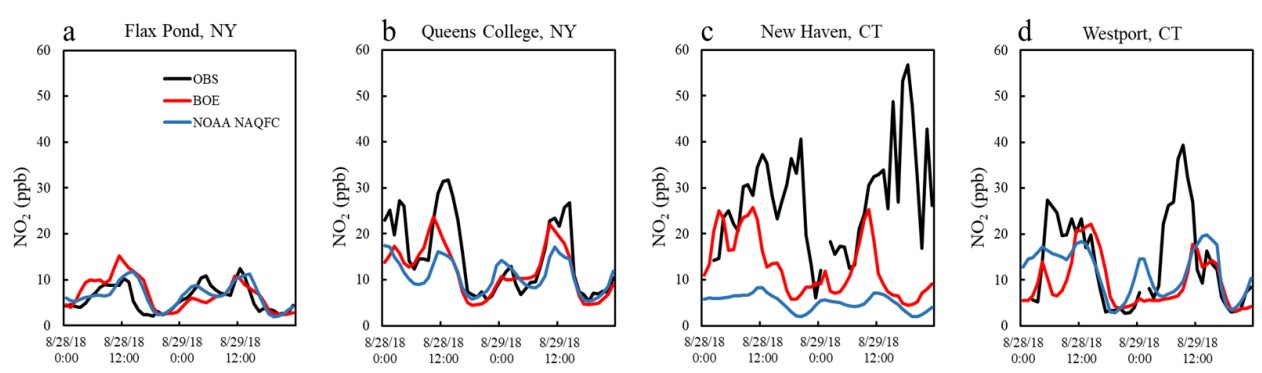


**Figure 10: Variations of observed (OBS) and simulated surface NO₂ concentrations by the 3km BOE system (BOE) and the 12km NOAA NAQFC system (NOAA NAQFC) at (a) Flax Pond, NY; (b) Queens College, NY; (c) New Haven, CT; and (d) Westport, CT sites during August 28–29, 2018.**

Next, the two model simulations are compared against the NO₂ VCD measured by NASA GCAS during the LISTOS field campaign. In order to allow a comparison between simulations and measurements from GCAS, the CMAQ prediction of NO₂ mixing ratio is vertically integrated from the surface to the layer which is the closest to the plane altitude to generate vertical column density (unit: molecules cm⁻²), with GCAS data averaged over the 3 km grid to provide a spatially representative observation data set. We also sample the model data to match the actual measurement time. The

GCAS observations show higher NO₂ VCD in the morning and lower values in the afternoon. This temporal pattern is well captured by both simulations. The GCAS observations depict an NO₂ hotspot over lower Manhattan and Brooklyn, which is reproduced by both BOE and NAQFC (Fig. 11). The observed and simulated VCDs are generally at the same magnitude (4–40×10¹⁵ molecules cm⁻²), with BOE better capturing the peak values. Moreover, the VCD prediction from the BOE run presents a northeastward pattern and it was lower over water area of LIS than that over surrounding lands.

In comparison, the VCD from NAQFC shows a high NO₂ plume over the land and the water around LIS. Compared to that from NAQFC, the spatial distribution of NO₂ VCD from BOE is more consistent with that of GCAS. This is also the case for the prediction of surface NO₂ distributions (Fig. 3), indicating the high-resolution system can outperform NAQFC through resolving the fine-scale processes. It should be noted that the VCD levels from both simulations are biased high outside the high emission density areas, especially in the morning. The BOE prediction shows a larger area of high NO₂

VCD than that from GCAS, suggesting either a positive bias in NOₓ emissions or inefficient transformation and removal of emitted NOₓ in the CMAQ model. The high NO₂ VCD from the NAQFC simulation is lower than the measurements over lower Manhattan and Brooklyn, and the high NO₂ VCD extends to an area larger than that from both GCAS and BOE. The performance is relatively unsatisfactory during the high pollution period in the morning of August 28 (Fig. 11e, 11i), with a correlation of only 0.56 for BOE and 0.44 for NAQFC. These low correlations could be in part caused by the

high spatial variability of fine resolution measured VCD, so that the averaged VCD is still more variable than either model. In contrast, the spatial patterns of NO₂ VCD in the afternoon are better reproduced than in the morning (Table S6). In addition, the NO₂ VCD from the simulation with combined adjustments using the EmisAdj_sub method for emission refresh shows a similar spatial pattern to that using BOE (Fig. S3). The NO₂ VCD level, however, is lower over the NYC

area suggesting an underestimate over the hotspot but much better prediction over the rest of the area. Besides the uncertainties in the model, an evaluation conducted by Judd et al. (2020) showed that the absolute difference in GCAS from Pandora measurements has an average and standard deviation of $-0.2 \times 10^{15} \pm 2 \times 10^{15}$ molecules cm$^{-2}$ and a percent difference on average of $-1.5\% \pm 20\%$. Overall, the BOE simulation at the 3 km resolution is able to reproduce the observed $NO_2$ VCD, and unlike the results of surface $NO_2$, the $NO_2$ VCD using EmisAdj_sub has lower NMB (33%) and NME (57%) compared to that using EmisAdj_avg (40% and 61%) while their correlation is still the same (0.74). It indicates the advantage of adjusting emission with a finer spatial resolution in simulating $NO_2$ vertical column in this study. Table S6 shows that both 3km simulations perform better than the 12 km NAQFC (R = 0.57, NMB = 45% and NME = 76%, respectively).

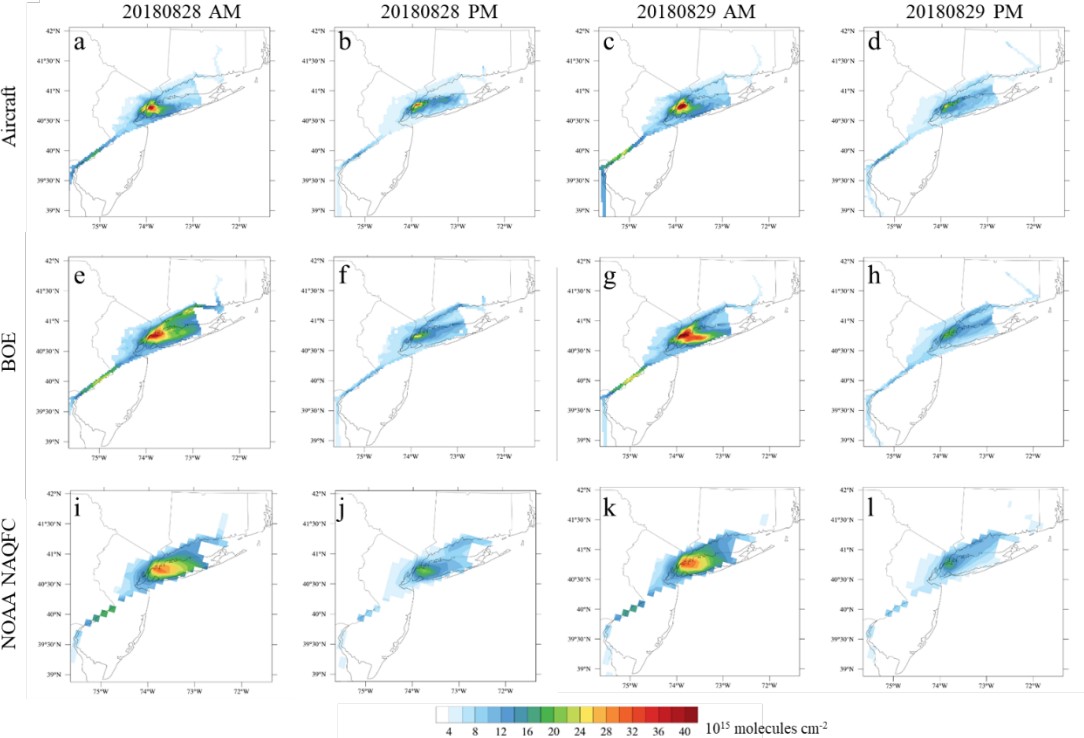

**Figure 11: Spatial distribution of NO₂ vertical column density (VCD) observed by NASA GeoCAPE Airborne Simulator (GCAS) (top row), and simulated by the 3 km BOE (center row) and 12 km NOAA NAQFC (bottom row) over the LIS domain during August 28–29, 2018. There were two flight missions each day: the morning flight (AM) from ~11:00 to 15:00 UTC and afternoon flight (PM) from ~16:00 to 20:00 UTC.**

### 4.2 O₃ prediction

One key result expected from the improved prediction system is better prediction of high O₃ episodes, especially those events that cause the exceedance of NAAQS. Here we compare the model performance between BOE and NAQFC at the seven sites where the O₃ concentrations exceeded the NAAQS. Compared to NAQFC, BOE demonstrates enhanced

prediction skills at all sites (Fig. 12). Note the comparisons may be attributed to the differences in meteorology, emission and other factors. Although it is difficult to attribute the improvement quantitatively to each factor, the magnitude of $O_3$ improvement from the base run to the BOE run is comparable to that of the overall reduced $O_3$ bias, suggesting a significant contribution from these improvement techniques. The results show that BOE can better capture peak $O_3$ values than NAQFC in the afternoon, a highly desired feature in predicting $O_3$ exceedances. Hourly surface $O_3$ concentrations

reached more than 100 ppbv at four Connecticut sites, including Greenwich, Westport, Middletown-CVH-Shed, and Stratford. While neither BOE nor NAQFC is able to predict such high values, BOE reduces the bias by 10-20 ppbv during peak hours at these sites. The improvement of peak $O_3$ prediction is less significant on the other sites with lower observed $O_3$ concentration, but BOE still displays better performance than NAQFC. There are only three sites at which one or both simulations overpredict peak $O_3$ on the August 29, 2018. Compared to NAQFC, BOE shows larger over-prediction of the

peak $O_3$ at the Greenwich site, but smaller overprediction at two other sites (Middletown and Westport).

        Besides better peak prediction, BOE has also improved the prediction of the timing of peak $O_3$. The peaks predicted by BOE are two to three hours earlier than that by NAQFC, which agree better with the timing of the observed peaks (Fig. 12). The BOE peaks are narrower than the NAQFC ones, so that the former follows the observed $O_3$ downslope and avoids the positive biases during late afternoon and early evening. Finally, BOE has improved the prediction of low $O_3$

concentrations and nighttime $O_3$ valleys that are lower than those from NAQFC. Both simulations, however, are unable to reproduce the extreme low nighttime values at several sites. Overall, the BOE simulation performs better in capturing daytime $O_3$ peaks and nighttime valleys, as well as the timing of both, with a mean correlation coefficient of 0.93 compared to 0.88 for the NAQFC simulation. This can be in part attributed to the high resolution of the LIS 3km system, which can better resolve meteorology and emission variations. As the emissions and meteorological inputs play an

important role in determining the magnitude and timing of high peaks (Pan et al., 2017), high resolution data of both emission and meteorology contributed to the improved the simulation of peak $O_3$ value and its timing, especially over urban areas (Figure 12).

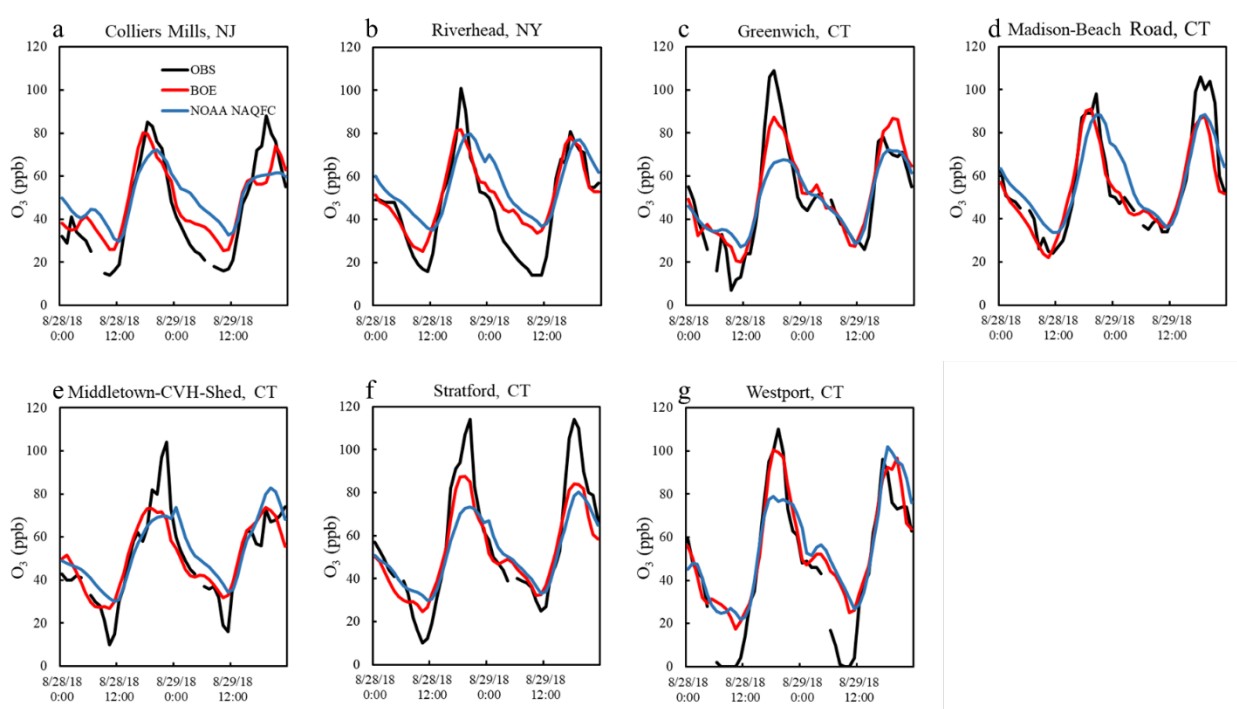

**Figure 12: Time series of observed (OBS) and simulated surface O₃ concentrations by the 3km system with dynamic Boundary conditions, OI initialization and Emission adjustment (BOE) and the 12km NOAA National Air Quality Forecast Capability System (NAQFC) system at the seven sites where the National Ambient Air Quality Standard (NAAQS) for O₃ were exceeded during August 28–29, 2018: a. Colliers Mills; b. Riverhead; c. Greenwich; d. Madison-Beach Road; e. Middletown-CVH-Shed; f. Stratford; and g. Westport.**

Vertical profiles of O₃ are compared between the Langley Mobile O₃ Lidar (LMOL) observations and the CMAQ simulations at the Westport site. As shown in Fig. 13, LMOL observations reveal that the O₃ concentration in the planetary boundary layer starts to build up around 16:00–17:00 UTC and high concentrations (>~70 ppbv), which extend to a height of about 1.5 km, last until 23:00 UTC on August 28 and 29. This pattern is reproduced by both the BOE and NAQFC simulations. Above the PBL, the variations of O₃ concentrations are also captured by both simulations. O₃ concentrations in the free troposphere are more controlled by regional O₃ production and transport than in the PBL. Consequently, the structure and magnitude of the O₃ profiles are very similar between the BOE and NAQFC simulations, since the BOE simulation is driven by the dynamic boundary conditions derived from the same NAQFC simulation. Compared to that from the LMOL observations, the predicted O₃ concentrations from both runs are biased low above 800 hPa but biased high below it. Between the two model simulations, the BOE run not only produces more O₃ in the PBL, but also shows a better temporal evolution of the PBL structure, with a short-lived high O₃ peak and a PBL height peak between 20:00–22:00 UTC on August 28, and persistent O₃ and PBL height plateaus between 16:00–23:00 UTC on August 29 (Fig. 13). The PBL in the BOE simulation extends well above 850 mbar, while the observed high O₃ from LMOL generally stays beneath this height, suggesting possible overprediction of the PBL height.

In general, the 3 km BOE simulation performs better to capture the temporal variability of the PBL and O₃ production but tends to overestimate both during this episode. In contrast, the NAQFC simulation has produced less pronounced

temporal variations in both $O_3$ concentrations and PBL height in the lower troposphere, in particular on August 28 when this region experienced the worst air quality in several states. The NAQFC simulation, however, performed better during the time with lower $O_3$ concentrations, which resulted in an overall lower NMB (9%) and NME (21%) comparing to that in BOE (22% and 26% respectively). The BOE simulation, however, presented a much better reproduction of the $O_3$ variability in term of correlation (0.71) than the NAQFC run (0.54). This suggests that the new 3 km BOE system is more responsive to the variations of the controlling factors that shape $O_3$ pollution, although the system needs to be further refined to reduce bias. The model performance for $O_3$ surface concentration and vertical distribution using the AFs from EmisAdj_sub is very close to those of using the AFs from EmisAdj_avg in the BOE case (Fig. S4, Table S7).

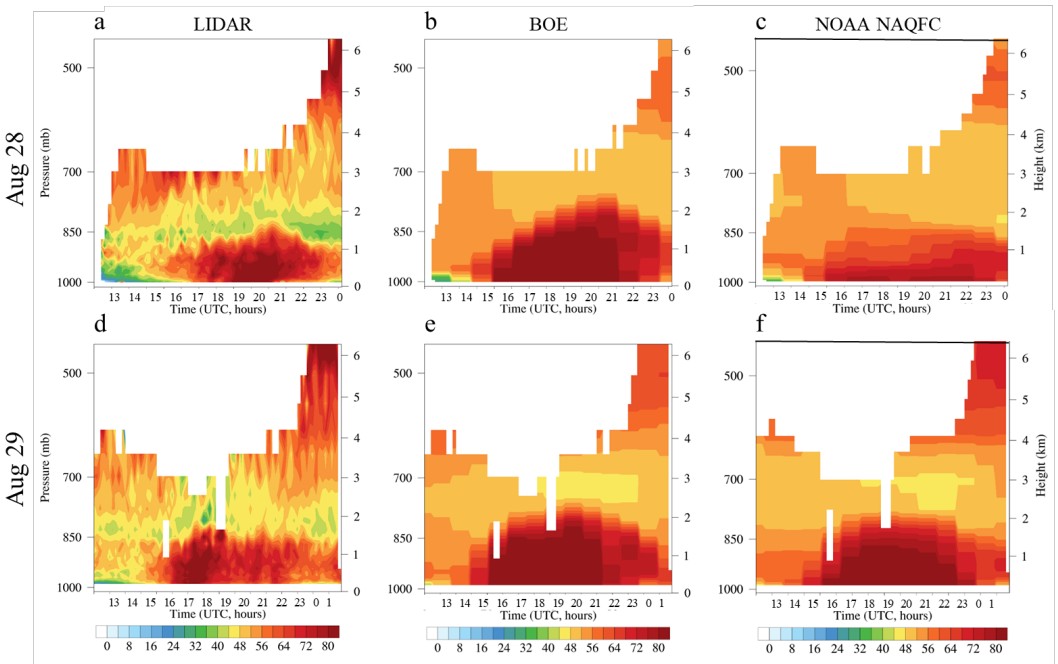

**Figure 13: Comparison of vertical $O_3$ profiles (observed by NASA Langley Mobile $O_3$ Lidar (left column, a & d) with these simulated by the 3 km prediction system (central column, b & e) and the 12 km NOAA NAQFC (right column, c & f) over the Westport site on August 28 (upper row) and August 29, 2018 (lower row), respectively. Note white represents missing data from the Lidar data.**

## 5. Summary

Improvement of air quality in the past decades renders the prediction of high ozone events more challenging. This study investigates the feasibility of designing a high-resolution air quality prediction system to capture these less frequent events with more accuracy. Relying on the observations collected during the Long Island Sound Tropospheric Ozone Study field campaign, we have assessed the effectiveness of various improvements to the prediction system to enhance the predictability of high $O_3$ episodes. These updates were then combined to explore how to further improve the predictability of both ozone and nitrogen dioxide. Finally, the modeling system with combined updates has been utilized to simulate a severe high $O_3$ pollution event in the Long Island Sound and surrounding areas.

Different prediction system updates demonstrate varying potentials to improve $O_3$ and $NO_2$ prediction performance. For $O_3$ prediction, the most significant improvement comes from the dynamic boundary conditions derived from NOAA National Air Quality Forecast Capability (NAQFC), compared to that with the static boundary conditions. This is due in part to the fact that the relatively small model domain used in this study and that $O_3$ is a regional air pollutant, making the its prediction more susceptible to the influence of regional transport. Dynamic boundary conditions (BCs) are less

influential for $NO_2$ prediction, for which all high-resolution simulations outperform the 12 km NAQFC simulation, highlighting the importance of spatially resolved emission and meteorology for the prediction of short-lived pollutants. The impact of improved initial concentrations through optimal interpolation (OI) is shown to be large in urban areas initially but fades away rapidly. The influence of OI adjustment, however, lingers for a longer period in an area with low emission density where emissions and chemical reactions make a smaller contribution to the $O_3$ budget than that in the

areas with high emission density. Such method may be more useful if applied to vertical layers above the ground. Future air quality forecasting and modeling can benefit from concerted efforts to provide near real time data of $O_3$ aloft on a continuous basis (Mathur et al., 2018), so that improved initialization of the aloft conditions can better represent regional transport and modulate the inferred impact of LBCs on $O_3$ prediction. Finally, emission adjustment, which changes the baseline emissions using the temporal trends derived from ground and satellite observations, only yields moderate

improvement in $O_3$ prediction compared to that without emission adjustment. One possible direction to explore is to apply other methods to constrain emissions that use both variational (*e.g.,* Elbern et al., 2007; Vira and Sofiev, 2012) and ensemble-based (*e.g.*, Miyazaki et al., 2012, 2017) solutions to analyze the 3D chemical tracers as well as their respective precursor emissions simultaneously. In addition, the importance of volatile consumer product VOCs has been identified in recent studies (McDonald et al., 2018), suggesting that updating other species than $NO_x$ is also necessary. This may be

challenging, however, through a similar approach to the NOx emission adjustment implemented here, since there are limited measurements of VOCs from both ground and space instruments. While the effectiveness of each update varies, a combination of these updates proves to outperform that with each single update. The new prediction system at 3 km resolution, equipped with dynamic BCs, OI and Emission adjustment (BOE), was used to simulate a high $O_3$ episode over the Long Island Sound region. Compared to the 12km operational NAQFC, the BOE system is able to significantly reduce

the biases in surface $O_3$ and $NO_2$ prediction. The BOE is also able to reproduce $NO_2$ VCD observed by NASA Langley GCAS with higher accuracy than the NAQFC. More importantly, the BOE simulation shows considerable improvement in capturing the $O_3$ peaks and valleys, as well as the timing of both, with a correlation coefficient of 0.93 compared to that of 0.88 by the NAQFC. This study demonstrates feasible measures to improve the capability of air quality prediction systems to capture high $O_3$ episodes in a cleaner urban environment.

*Data Availability.* WRF is an open-source community model. The source code is available at http://www2.mmm.ucar.edu/wrf/users/download/get_source.html (last access: September 2021). CMAQ and SMOKE source code is available on the Community Modeling and Analysis System (CMAS) Center of University of North

Carolina, Chapel Hill: https://www.cmascenter.org/ (last access: July 31, 2021). The AirNow hourly data of $O_3$ and $NO_x$ are available at https://files.airnowtech.org/?prefix=airnow/ (last access: May 2021) and the hourly $NO_x$ data from US

EPA Air Quality System (AQS) surface network is available at https://aqs.epa.gov/aqsweb/airdata/download_files.html (last access: May 2021). The GCAS $NO_2$ vertical column density and the LMOL $O_3$ vertical profile data are available at https://www-air.larc.nasa.gov/missions/listos/index.html (last access: May 2021). The monthly product of $NO_2$ vertical column density from OMI is available at https://avdc.gsfc.nasa.gov/pub/data/satellite/Aura/OMI (last access: July 31, 2021).

*Author Contribution*. DT and SM designed the study, conducted the simulations and wrote the manuscript. JW, XZ and PL helped development of the modeling system. LL, RS and LJ provided OMI and LISTOS field campaign data and helped interpreting the results. YT and TC provided code for the original OI method. All authors edited and commented on the manuscript. All authors read, revised, and approved the final paper.

*Competing interests*. The authors declare that they have no conflict of interest.

*Acknowledgement.* This work was partially supported by a National Research Council fellowship to S. Ma at NOAA Air Resources Laboratory and by NOAA Weather Program Office and Robert Wood Johnson Foundation to D. Tong. The authors are grateful to the EPA and NYDEC for sharing the AQS data and to NASA for providing the OMI, GCAS and Langley Mobile $O_3$ Lidar datasets.  Finally, we want to thank the Editor for handling our submission and two anonymous reviewers for their constructive comments on earlier versions of this paper.

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
