# Peer review of "Improving predictability of high ozone episodes through dynamic boundary conditions, emission refresh and chemical data assimilation during the Long Island Sound Tropospheric Ozone Study (LISTOS) field campaign"

_Atmospheric Chemistry and Physics, 2020_

## Referee Comment (RC2)

[referee-annotated manuscript omitted]

---

## Author Response (AR1)

**Responses to Reviewer Comment #1:**

We want to thank the reviewer for his/her detailed and thoughtful comments. Our point-to-point responses to these comments are provided below, with proper changes made in the revised manuscript.

*1. At many places in the manuscript it is sated that a high-resolution forecast system has been developed for the LIS region. Is this an operational forecast system? Was the system operational for the entire LISTOS study? What relation does it have to the NAQFC system which is also often mentioned? Were the base and additional simulations conducted in forecast or retrospective mode? Did the WRF simulation employ data assimilation – if so, was the assimilation strategy as one would use it in forecast mode? What was the extent of the WRF modeling domain? Was WRF deployed in a nested mode – is so what was the extent of the outer domain and how did it compare with the meteorology used for the NAQFC which provided the chemical LBC for the high-resolution domain?*

Response: The NAQFC is the operational air quality forecasting system operated by the National Oceanic and Atmospheric Administration (NOAA), while the LIS system presented in this study is a new research air quality prediction system under development. It is not operational. We have added this information to avoid confusion (see Lines 95 & 146 in the revised manuscript).

The LIS study has no direct relationship with NAQFC, although it may provide some useful information for NAQFC to develop a future high resolution forecasting system with emission and chemical data assimilation capabilities. NAQFC is mentioned here because it is the national operational forecasting system, hence being used as a benchmark to evaluate the model performance of the LIS system. To help understand the difference between NAQFC and LIS, we added a table to the Supplemental Information (SI) as shown below:

Table 1. Comparisons between LIS and NAQFC configurations during the LISLOS study.

| Configuration | LIS | NAQFC |
|---|---|---|
| Horizontal resolution | 3km | 12km |
| Meteorology | WRF with Global Forecasting System (GFS) acting as ICs/BCs | North America Mesoscale Forecast System (NAM) |
| Lateral Boundary conditions | Various (Default/NAQFC) | climatological gaseous LBC from GEOS-chem. Dynamic aerosol LBC from GEFS-Aerosol |
| Initial concentration | CMAQ restart file | Previous run |
| Base Emission | NEI2011v2 | NEI2014v2 |
| Chemistry | CB6 Aero6 | CB05 Aero6 |

Both the base run and adjustment simulations were conducted in forecast mode. The $O_3$ episodes during the LISTOS field campaign were chosen here because of the rich pool of measurements (aircrafts, lidar, etc.) that can be used for detailed model evaluation.

Regarding data assimilation for meteorology, it was not applied in the WRF model used in this study, as we primarily focused on improving the performance of the CMAQ model. The WRF model was not deployed in nested mode and its domain was one grid larger on each boundary compared to that of the CMAQ model. The WRF model run, however, was driven with initial and boundary conditions from the NOAA NCEP's Global Forecast System (GFS) simulation, which was equipped with various data assimilation techniques. We have clarified this issue in the manuscript and added the description of the configuration in Lines 151-153:

"No data assimilation was applied in the WRF simulation. The model is conducted in a single domain with 132×122 grid cells with one grid more on each boundary compared to that of the chemical transport model."

The meteorology data from NAQFC and LIS could not be compared to each other since NAQFC, in its public release, only provides CMAQ output files, not the input files, such as meteorology and emissions. Hence, such a comparison is not feasible and beyond the scope of this study. We acknowledge that many factors listed in the Table above can contribute to the difference in model performance, including meteorology inputs. As NAQFC is driven by operational weather forecasts, it is not surprising that the WRF simulation may not be as great in the NOAA operational weather forecast. Nevertheless, the new system performs better even with the current WRF input, highlighting the importance of other factors such as model resolution, emissions and chemical data assimilation, in improving prediction performance.

*2. If the high-resolution forecast system has been deployed for an extended period, how does its general performance compare to that for the period of focus of this study, i.e., August 25-31, 2018? This aspect is specifically important to understand if there were conditions unique to the high ozone episode examined here or model attributes that led to the noted performance characteristics. On the other hand, if this is a limited modeling study that was conducted to examine this specific ozone episode it is okay to state that so the results can be viewed and assessed in the right context.*

Response: Thank you. This is a good point. As mentioned earlier, this is a research prediction system, not operated continuously. We have clarified it in the revised manuscript (L146-147):

"The high-resolution air quality forecasting system used here is a new research prediction system deployed during the 2018 LISTOS field campaign period."

In addition, we have conducted an extended model simulation beyond the LISTOS field campaign period, and presented the model evaluation with observations from the ground monitoring network AQS. During the cold season, the simulated $O_3$ showed similar concentration level with the observations (see examples in Fig. 1a), as well as the spatial patterns (i.e., on Jan 15, 2020, the domain-average correlation coefficient (CORR) was 0.76 and RMSE was 6.78 ppb). In the summer season, the simulation without adjustment showed acceptable correlation (CORR=0.92) but it underestimated the concentration (Fig. 1b). With the adjustment, the model bias was reduced and the predicted concentrations were brought closer to the observed levels (Fig. 1c). Its CORR in NYC could reach up to 0.99. The model shows relatively weaker performance in the springtime, such as the average CORR of 0.45 on March 14, 2020. It means this system is able to forecast the $O_3$ activities over this region in most time and these adjustments could improve the performance of daily run, but it still needs further improvement.

[Figure]

Figure 1. Diurnal variation of observed (blue line) and simulate (red line) $O_3$ on Jan 15 (a), May 20 (b), June 20 (c) and March 14 (d) 2020.

*3. It is not terribly surprising that for the limited geographic extent of the modeling domain considered here, the specification of chemical lateral boundary conditions is found to be influential on predicted ozone (and not $NO_x$ but I suspect also for many other species such CO, $PM_{2.5}$ etc.) distributions. Given lifetimes of these species and the typical advective time-scales this should be an expected outcome. It is not readily apparent what constitutes the default static LBC profile – what were the typical values for $O_3$ and other species examined? Looking at the CMAQ documentation, it appears that the default LBC profile provided represents "clean" tropospheric conditions and is recommended for use along model boundaries that are typically over remote regions devoid of significant emission forcing. From the model documentation it also appears that even those are now often substituted with conditions derived from hemispheric versions of the model. It is thus a bit surprising that for a "new" high-resolution forecast system over a high emission density region such as the northeast corridor one would consider such a profile and not a nested configuration to capture the space and time varying chemical conditions of air masses advected to the LIS. In light of this, the statement on L265 "This suggests the default profiles provided by CMAQ represent a clean environment, such as marine air layer, and are not suitable for areas with active emissions and tropospheric $O_3$ production" is somewhat trivial. Clearly, such impacts are well recognized as the authors do attempt to account for such by using data from the NAQFC. A clearer description of the model set up and reasons for not using a consistent one-way nested configuration (with consistent treatment of meteorology, emissions, initialization for the outer domain) to better capture the LBCs would be useful.*

    Response: Thank you for the suggestion. Indeed, the default static LBC profile represents clean air environment. We provide in Table 2 below the different values of $O_3$, $NO_2$, CO and sulfate (PSO4J, as an example of PM species) in the dynamic BC files from NAQFC and that from the default profile. Between the two LBCs, the $O_3$ concentrations increased from 37.66 ppb to 51.28

ppb, but the concentrations of $NO_2$ did not vary considerably during the episode. The concentrations in profile BCs are 35.0 ppb ($O_3$), 0.07 ppb ($NO_2$), 75.95 (CO) and 1.134 µg m$^{-3}$ (PSO4J), respectively. The differences in concentration between the profile and the NAQFC dynamic BCs were higher in the summer (high $O_3$) season while low in cold season. It was consistent with the daily simulation results shown in Fig. 1 (see response to the comment above).

Table 1. Averaged concentrations of $O_3$, $NO_2$, CO (unit: ppb) and PSO4J (unit: µg m$^{-3}$) in the dynamic/static LBC files

|  | Date | $O_3$ | $NO_2$ | CO | PSO4J |
|---|---|---|---|---|---|
|  | 20200825 | 38.04 | 1.56 | 112.85 | 2.545 |
|  | 20200826 | 37.66 | 1.42 | 113.49 | 1.944 |
| Dynamic LBCs | 20200827 | 44.12 | 1.54 | 133.96 | 3.445 |
|  | 20200828 | 48.07 | 1.88 | 152.18 | 4.642 |
|  | 20200829 | 51.28 | 1.65 | 155.05 | 4.721 |
|  | 20200830 | 46.63 | 1.52 | 133.08 | 2.921 |
| Static LBCs |  | 35.0 | 0.07 | 75.95 | 1.134 |

The purpose of including the profile BCs was not to suggest using it in the final simulation, but as the control case from which we can evaluate and quantify the effectiveness of different adjustment methods in the forecasting systems. The underestimated $O_3$ in the control run also indicates the importance of dynamic BC during high pollution time and the influence of transported pollutants on the LIS region. In the revised manuscript, we have clarified the role of the simulation with the profile BC in the simulation design in Sect. 2.1 (L129):

"The first group (Control run) applies no adjustment, using default profile as LBCs. It serves as the reference case to allow quantifying the effectiveness of each adjustment method."

Furthermore, we removed the statement on L265 in the last version as suggested by the reviewer, and added the discussion of the influence of dynamic BCs on different season and pollutants in L304:

"As the profile BCs are static and lack spatial-temporal variations, the Control run mainly reflects the local contributions of emissions, transport and chemical processes within the domain (Tang et al., 2007). The underprediction suggests that these processes are insufficient to produce the observed $O_3$ levels, and that the transport of air pollutants from upwind is important to predict the high $O_3$ episodes. It highlights the significant influence of dynamic BCs on the simulations over this region during high pollution time. In comparison, the influence of BCs is less important

during the cold season, when the simulation with the profile BCs can also result in prediction in reasonable agreement with observations (Fig. S2a, d)."

And more discussion in L340:

"In addition, the improvement of simulated $NO_2$ using dynamic BC was much smaller compared to that of $O_3$. This is because the lifetime of $NO_2$ is relatively short (1–7 h in summertime, Lu et al., 2015), and its budget in urban areas is mainly influenced by local emissions and chemistry, and less by regional transport, indicating the effectiveness of dynamic BCs depends not only on the downwind/upwind gradients, but also on lifetimes of the concerned species."

Previous studies on regional $O_3$ modeling usually used nested model to generate the BCs for the inner target domain area (such as Taghavi et al., 2004; Fu et al., 2009; Yin et al., 2015). The objective of this study is to set up a forecasting system that can be deployed fast during field campaigns. The nested models will need higher computational resources and a longer execution time. So here we used the NAQFC product as dynamic BCs directly and then tested the performance of such a system. The results show that, even without the nested configuration, the results are in good agreement with observations and we can apply this for the real-time forecast run. We added the explanation why choosing NAQFC as BCs instead of using a nested modeling system in Sect. 2.3 (L185-191):

"In the previous studies of regional modeling, a nested grid approach was often applied to provide dynamic BCs for the study area (e.g., Taghavi et al., 2004; Fu et al., 2009; Yin et al., 2015). However, the nested model would need higher computational resources and a longer running time. The increasing pool of real-time national and global forecasts provides alternative BCs that be used to drive a regional forecasting system as demonstrated in this work. Here, we explore the feasibility of utilizing the products of NOAA NAQFC, which provides real-time national forecasts to prepare dynamic boundary conditions to drive the LIS3km system. The NAQFC is an operational system, operated by the National Weather Services, and the data are provided freely to the public."

Reference:
Fu, J. S., Streets, D. G., Jang, C. J., Kebin He, J. H., He, K., Wang, L., Zhang, Q. Modeling regional/urban ozone and particulate matter in Beijing, China. Journal of the Air & Waste Management Association, 59(1), 37-44, 2009.
Taghavi, M., Cautenet, S., and Foret, G.: Simulation of ozone production in a complex circulation region using nested grids, Atmos. Chem. Phys., 4, 825–838, https://doi.org/10.5194/acp-4-825-2004, 2004.
Yin, S., Zheng, J., Lu, Q., Yuan, Z., Huang, Z., Zhong, L., Lin, H. A refined 2010-based VOC emission inventory and its improvement on modeling regional ozone in the Pearl River Delta Region, China. Science of the Total Environment, 514, 426-438, 2015.

*4. Perhaps one positive consequence of using the default LBC and comparisons with the run with the NAQFC LBC is demonstrating the expected influence of regional transport on $O_3$ levels in the LIS region, with a suggested enhancement of 10-20 ppb in peak hourly ozone on different days due to influences from outside the modeling domain, assuming that the discrepancies relative to the observations can be solely attributed to LBC and not other model processes or input? How representative is the limited set of conditions modeled here of the high ozone days in the LIS region? There were several days during summer 2018, outside the period examined here, when the LIS region witnessed high ozone levels.*

Response: We appreciated this thoughtful suggestion. Indeed, the difference between the default LBCs (background levels) and the dynamic LBCs from NAQFC represents the influence of regional transport on $O_3$ levels. Although the reasons that cause the model bias are complicated, the pollutant differences between Control and BCON run can be generally attributed to the pollutant transport, as the only difference between these runs is the LBC input. According to the EPA (Fig. 2), high $O_3$ levels were observed throughout the northeast corridor during the August 29, 2018 $O_3$ episode, with the highest values in the LIS region. The high $O_3$ concentrations in LIS were contributed by both local production and regional transport, and the transported source might contribute significantly during the peak time. We made this explanation in Sect. 3.1 (L304):

"As the profile BCs are static and lack spatial-temporal variations, the Control run mainly reflects the local contributions of emissions, transport and chemical processes within the domain (Tang et al., 2007). The underprediction suggests that these processes are insufficient to produce the observed $O_3$ levels, and that the transport of air pollutants from upwind is important to predict the high $O_3$ episodes. It highlights the significant influence of dynamic BCs on the simulations over this region during high pollution time. In comparison, the influence of BCs is less important during the cold season, when the simulation with the profile BCs can also result in prediction in reasonable agreement with observations (Fig. S2a, d). This indicates the influence of dynamic BCs varies with time and it is more significant during the high pollution time."

[Figure]

Figure 2. Ozone AQI by site on August 29, 2018 (source: https://www.epa.gov/outdoor-air-quality-data/air-data-concentration-map).

Regarding the representativeness of the simulated episodes, we have conducted additional simulations for an extended period (see the response to Comment #2). the system demonstrates a stable performance for the time outside the LISTOS period.

*5. One interesting aspect depicted in Figure 2 is that the peak $O_3$ simulated in the control run is nearly the same on all days, suggesting that at least for the limited number of days examined in this study, emissions within the domain have about the same daily impact on average ozone. Assuming that the emissions within the domain are captured correctly and the WRF simulations represent the prevalent meteorology (neither of which are necessarily assessed), comparisons with the NAQFC and dynamic LBC case are suggestive of relatively large regional contributions on*

*these days – how representative are these regional contributions of high O$_3$ in the LIS region? This also links back to my earlier comment on the skill of the new forecast system over an extended time-period.*

Response: This is a good point. We have added, with thanks, the following statement in the revised manuscript to reflect this observation (L300):

"Note the peak O$_3$ simulated in the control run is nearly the same on all days during the simulation period. The comparisons between the peak O$_3$ with the default profile and dynamic LBC case indicates relatively large regional contributions on these days."

We have addressed the representativeness issue above.

*6. The description of the optimal interpolation application would benefit from additional clarity. If I understand the methodology, surface observations are used to adjust the model initial state for O$_3$ and perhaps NO$_x$ (please state that explicitly if that is the case). However, it is not readily apparent if this is done every 24 hours? At what frequency is the OI applied? Also based on the strong forcing at the surface from emissions and deposition, it is to be expected that the influence of the OI fade away rapidly. It appears that such methods may be more useful to aloft data (e.g., https://doi.org/10.1021/acs.est.8b02496). Did the authors consider using aloft observations from the O$_3$ lidar in conjunction with the OI to explore possible improvements in short-term air quality forecasts? Improved initialization of the aloft conditions may also help better represent regional transport and modulate the inferred impact of LBC specification.*

Response: We have provided additional information of the OI application to address the issues raised here. In this study we use OI to adjust O$_3$, NO$_2$ and NO in the initial conditions files, namely the restart (CGRID) file in CMAQ. We added the following sentences in Sect. 2.3 (L202-204):

"In the CMAQ model, the restart file, called CGRID, is daily generated during the simulation and acts as ICs for the next day. To constrain the biases in ICs, the concentrations of ozone, NO$_2$ and NO in the restart file were adjusted via the OI method, which is applied every 24 hours at 0:00 Coordinated Universal Time (UTC)."

We agree with the suggestion to utilize more aloft measurements in the OI application. The O$_3$ lidar data from NASA LMOL are not available routinely in near-real-time (NRT), so it is not an option to be used for air quality forecasting. It was used here to evaluate CMAQ prediction of O$_3$ profiles. As suggested by Mathur et al. (2018), future air quality forecasting and analysis modeling can benefit from concerted efforts to provide NRT data of ozone aloft on a continuous basis to quantify its contribution to ground-level concentration. We add the following statement in the revised manuscript (see Lines 689-696 in Section 5. Summary):

"The impact of improved initial concentrations through optimal interpolation (OI) is shown to be large in urban areas initially but fades away rapidly. The influence of OI adjustment, however, lingers for a longer period in an area with low emission density where emissions and chemical reactions make a smaller contribution to the O$_3$ budget than that in the area with high emission density. Such method may be more useful if applied to vertical layers above the ground. Future air quality forecasting and modeling can benefit from concerted efforts to provide near real time data of O$_3$ aloft on a continuous basis (Mathur et al., 2018), so that improved initialization of the aloft conditions can better represent regional transport and modulate the inferred impact of LBCs on O$_3$ forecasting."

Reference:

Mathur, R., C. Hogrefe, A. Hakami, S. Zhao, J. Szykman, and G. Hagler.: A call for an aloft air quality monitoring network: need, feasibility, and potential value. Environ. Sci. Technol, 52 (19), 10903–10908, 2018.

*7. The combined impacts of the LBC and emission adjustment simulations raise an interesting point on the representativeness of the NAQFC derived LBCs for the "forecast" year. Based on the arguments put forth, it appears that the "new" emission adjustments were applied only to the 3km resolution domain extent. By inference, the emissions utilized in the NAQFC are likely biased high since they did not benefit from these adjustments in the inferred emission trajectory since the NEI year. Conceivably, reducing regional $NO_x$ emissions outside of the study domain will reduce the regional transported $O_3$ to the study region thereby possibly increasing the already low bias for the high $O_3$. What emissions were used in the NAQFC runs – did they benefit from similar "refresh" (reductions) relative to the NEI as the those within the LIS domain? Please clarify these aspects and any possible effects associated with a high bias in regional $O_3$ from NAQFC arising from possible high bias in emissions used in the NAQFC.*

Response: The emission refresh was not implemented in NAQFC, which uses the 2014 NEIs for anthropogenic emissions. Without the emission reduction, the changes in $NO_x$ emissions between the inventory and forecast years are not accounted for. Therefore, an over-prediction is expected of all other model inputs are well represented in the model. Due to the biases in other model inputs, such as meteorology and chemistry, $O_3$ prediction can be either over-predicted or under-prediction, depending on the actual combination of these factors (e.g., see Fig. 2c and 2d in the manuscript). On the high $O_3$ days, NAQFC over-predicted surface $O_3$ (Fig. 2c). Using the NAQFC LBCs, the BCON LIS simulation overpredicted $O_3$ during peak hours (Fig 2d), although it is difficult to attribute all bias to NAQFC emission bias without running a sensitivity test with the NAQFC system, which is beyond the scope of this study.

We have clarified the aspects of NAQFC emission data, and discussed the potential effect associated with the unadjusted $NO_x$ emissions on LBCs and LIS $O_3$ prediction (Lines 460-465 in Section 3.3):

"Note that the emission adjustment was only implemented in the LIS system, not in NAQFC, which still uses the 2014 NEIs for anthropogenic emissions. Without the emission adjustment, the changes in $NO_x$ emissions between the inventory and forecast years are not accounted for. On the high $O_3$ days, NAQFC over-predicted surface $O_3$ during the study period (Fig. 2c). The NAQFC LBCs are likely associated with a possible over-prediction of the regional transport, which can be partially responsible for the BCON LIS simulation overpredicted $O_3$ during high $O_3$ days (Fig 2d)."

Reference:
Tang, Y., Bian, H., Tao, Z., Oman, L. D., Tong, D., Lee, P., Campbell, P. C., Baker, B., Lu, C.-H., Pan, L., Wang, J., McQueen, J., and Stajner, I.: Comparison of chemical lateral boundary conditions for air quality predictions over the contiguous United States during pollutant intrusion events, Atmos. Chem. Phys., 21, 2527–2550, https://doi.org/10.5194/acp-21-2527-2021, 2021.

*8. Even though the study promotes high resolution modeling, much of the analysis focuses on aggregate metrics (averaged over model-station pairs). Thus, the relative advantages of using the 3km resolution are not readily apparent. Some discussion of the gains realized from higher resolution (say even relative to the 12km NAQFC) would be useful.*

Response: Thank you. We have added additional discussion of the gains realized from higher resolution in the revised manuscript:

In Section 3.1 (L322-327):

"Compared to that in the NAQFC prediction, the $O_3$ from the 3 km BCON run demonstrated more detailed spatial distributions in the predicted $O_3$ fields. For instance, the $O_3$ concentration over the Long Island Sound is lower than its surroundings and the 3 km simulation could reproduce this pattern while the $O_3$ from the 12 km NAQFC showed a relatively coarser pattern of the concentration gradient. The $O_3$ distribution along the coastal area also agrees better with the observations than the 12 km NAQFC prediction. This proves the high-resolution simulation can better reproduce the pollutant variability over this coastal urban area."

In Sect. 4.1 ($NO_2$ VCD distribution), L581-586:

"Moreover, the VCD prediction from the BOE run presents a northeastward pattern and it was lower over water area of LIS than that over surrounding lands. In comparison, the VCD from NAQFC shows a high $NO_2$ plume over the land and the water around LIS. This spatial distribution from BOE is more consistent with that of GCAS compared to that from NAQFC. Similarly, this situation is also similar for the prediction of surface $NO_2$ distributions (Fig. 3), indicating the high-resolution system can outperform NAQFC through resolving the fine-scale processes."

In addition, we compared the hourly variations of observed and simulated $O_3$ and $NO_2$ at individual monitoring sites. We found that the results from the 3 km system showed better performance, especially in predicting the peak values and timing of $O_3$ concentrations during the exceedance the NAAQS. This may be partly attributed to the high resolution of the simulation. So we added the analyses in L637:

"As the emissions and meteorological inputs play important roles in determining the magnitude and timing of high peaks (Pan et al., 2017), the emissions and meteorological data at the 3 km resolution could improve the simulation of peak values and timing, especially over urban areas."

*9. L76-77: "complex urban areas" is a vague – please elaborate on the specific challenges for air quality forecasts.*

Response: We have added additional information to elaborate on the specific challenges (L75): "Prior studies have also revealed that air quality models face additional challenges in predicting surface $O_3$ concentrations at coastal locations or over complex urban areas, including uncertainties in vertical mixing, deposition processes, spatial-temporal allocation of emissions to the air quality models (Hogrefe et al., 2007; Tong et al., 2006)."

*10. L82: perhaps should say six different sources or six different representations of lateral boundary conditions.*

Response: Thank you. We have changed it to "examined the impact of six different sources of lateral BCs" (L82)

*11. L93: Is "modeling techniques" the correct terminology for the experiments that largely investigate different forms of input data to the CTM? I wouldn't necessarily characterize changes in input data (initial conditions, boundary conditions, and emissions) as modeling techniques, especially since the methodology (at least the emission adjustment and OI) appear to be based on previously investigated methods.*

Response: These experiments include both the input data, and the underlying tools/techniques to prepare these data. Although this study focuses on improving input data to the model, as the

reviewer pointed out, each input was prepared by a tool or an algorithm, which was commonly referred to as modeling techniques, such as Tang et al. (2017) that called OI used to improve initial conditions as a chemical data assimilation technique. Admittedly, these techniques are not applied to alter the CTM, the core of the air quality forecasting system, but to alter the inputs, which are also parts of the overall modeling system in a broad sense.

Reference:
Tang, Y.H., Pagowski, M., Chai, T.F., Pan, L., Lee, P., Baker, B., Kumar, R., Delle Monache, L., Tong, D., Kim, H.-C., A case study of aerosol data assimilation with the Community Multi-scale Air Quality Model over the contiguous United States using 3D-Var and optimal interpolation methods, Geosci. Model Dev., 10, 4743–4758, https://doi.org/10.5194/gmd-10-4743-2017, 2017.

*12. L111: It's not clear to me what does domain size has to do with fine-scale processes - I would have thought grid resolution would be more influential in that regard – perhaps this sentence would benefit from some restructuring.*
    Response: Thank you. We have rephrased this sentence to avoid confusion (L113):
    "While this model domain is large enough to capture key physical/chemical processes within the LIS area, such as sea breeze circulation and photochemistry, the influence of regional transport outside this domain cannot be adequately represented."

*13. L181: it was not apparent to me what the 11×11 grid cell block refers to and what its relevance is? If it's an area of influence in the OI, please provide some rationale for its choice?*
    Response: Yes, the 11×11 grid cell block refers to the area of influence in the OI. This area of influence choice is based on correlation length analysis. According to the calculations of correlation length shown in Chai et al. (2017), it showed a correlation length scale of ~84 km, which is the separation distance where the corresponding correlation coefficient falls to $e^{-1}$ when using the National Meteorological Center (NMC) approach, while the Hollingsworth-Lönnberg method has a longer correlation length scale (~160 km). Chai et al. (2017) chose 84 km as the OI influence range for the CONUS domain. Moreover, this influence length scale also varies from region to region. Over remote regions, the length scale may be longer and it should be shorter over polluted areas as the correlation reduces more rapidly. Considering the high emission density and high model resolution over the LIS area, we chose a shorter influence length (33km) for a higher correlation threshold (r >= 0.5) for the LIS area. We have provided this explanation in the revised manuscript (L204).

Reference:
Chai, T., Kim, H. C., Pan, L., Lee, P., and Tong, D.: Impact of Moderate Resolution Imaging Spectroradiometer aerosol optical depth and AirNow PM2.5 assimilation on Community Multi-scale Air Quality (CMAQ) aerosol predictions over the contiguous United States, J. Geophys. Res.-Atmos., 122, 5399–5415, https://doi.org/10.1002/2016JD026295, 2017.

*14. Was the emission "refresh" applied only to $NO_x$ emissions or were emissions of other species also modulated relative to the base NEI. One would imaging that VOC emissions would have also changed between 2011-2018. Please also clarify why the 2011 NEI (L200) is used when*

*conceivably updated (and closer to the forecast year) versions of the NEI (2014, 2017) may have been available?*

Response: The emission refresh was applied only to $NO_x$ emissions, not to the emissions of other species. VOC emissions may have changed as well, but the emission sources are different for VOCs and $NO_x$. Previous studies revealed that decrease in $NO_x$ emissions has shifted the transition zone of $NO_x$-saturated to $NO_x$-limited regimes closer to urban centers and it showed increasingly $NO_x$-limited ozone chemistry in warm seasons in the NYC area (Jin et al., 2017; Jin et al., 2020). This means the regional $O_3$ production is more controlled by $NO_x$ emissions than VOCs. Finally, unlike $NO_x$, biogenic sources contribute a large portion of the VOCs emissions during summertime. In this study, the biogenic VOC emissions are calculated inline, so the biogenic VOC emissions are updated using real-time meteorology.

When we started to prepare this forecasting system, the 2014NEI had not been finalized and well tested. Therefore, we chose the well vetted 2011NEI version 2 to provide the emission input file. For this study, the outdated NEIs also provided the opportunity to apply and evaluate the emission adjustment approach to using the observed trends from NEI years to the forecast year.

Reference

Jin, X. M., Fiore, A. M., Murray, L. T., Valin, L. C., Lamsal, L. N., Duncan, B., Boersma, K. F., De Smedt, I., Abad, G. G., Chance, K., and Tonnesen, G. S.: Evaluating a space-based Indicator of surface ozone-NOx-VOC sensitivity over midlatitude source regions and application to decadal trends, J. Geophys. Res.-Atmos., 122, 10231–10253, 2017.

Jin, X., Fiore, A., Boersma, K. F., Smedt, I. D. and Valin, L., Inferring Changes in Summertime Surface Ozone–NOx–VOC Chemistry over US Urban Areas from Two Decades of Satellite and Ground-Based Observations. Environ. Sci. Technol., 54(11), 6518-6529, https://doi.org/10.1021/acs.est.9b07785, 2020.

15. L150: Were model estimates of isoprene concentrations compared with observations from the LISTOS study? What may be the possible role of uncertainties in isoprene emissions within the LIS region on model predicted $O_3$ and its discrepancies relative to observations?

Response: We have added new comparisons of the predicted and observed isoprene concentrations (see the figure below). There were limited measurements of isoprene in the LIS region, and the field campaign for VOC sampling occurred outside the study period. Instead, we compared the simulated hourly isoprene with AQS observations from a monitoring site in Bronx, NYC (Fig. 4). The predicted isoprene concentration agrees well with the observations regarding both levels and diurnal patterns, except underpredicting the peak values. The correlation coefficient is 0.93, higher than that of $NO_2$. While the evaluation is limited, the results indicate that the role of VOC uncertainties may not be a major concern, although future study is guaranteed to look into this issue with more details. We added the figure below and relevant discussion in the Supplemental Information (see Figure S5).

[Figure]

Figure 4. Observed and simulated hourly isoprene concentrations at Bronx, New York (40.868°N, 73.878°W) during the episode.

*16. Figs 2d and 3d - please state the time zone on the x-axis label - looks like UTC?*
    Response: The time zone (UTC) has been added to the x-axis.

*17. L310-312: Please restate here and indicate in Table 2 caption the length over which these metrics are computed - are these for hourly paired model and observations.*
    Response: The averaging length is hourly for the period of August 26 to 31, 2018. We have added this information to Table 2 and revised the text (L373):
    Table 2 caption: "Regional mean statistical metrics between hourly observed and simulated $O_3$ from August 26 to 31, 2018 over the Long Island Sound region".
    New L376: "The statistical metrics calculated from hourly simulated and observed data from August 26 to 31, 2018 were reported in Table 2, the RMSEs for $O_3$ are reduced from 14.97 ppbv to 13.72 ppbv in the OI_bias run, to 13.79 ppbv in the OI_idw run and to 14.30 in the OI_avg run."

*18. L323-324: This is somewhat of a misleading statement - the spike is just an indication of the model error at a specific time and location - not necessarily the impact of OI in large metropolitan areas. Please reword this sentence.*
    Response: Thank you. We have reworded the sentences as follows (L400):
    "In large metropolitan areas, OI adjustments result in spikes in large metropolitan areas indicate the model errors at the time of OI adjustment at the monitor sites, with the mean errors being up to 14 ppbv in surface hourly $O_3$ concentrations over NYC and 16 ppbv over Philadelphia, respectively."

*19. L325-327: I find the suggestion that the magnitude of the OI adjustment is related to the emission strength/density to be speculative and not substantiated by any presented analysis. Restructuring the discussion would be useful.*

Response: Thank you. We revised the sentence and have deleted the speculative part of the statement (L404):

"The New Haven–Hartford region sees a smaller change of $O_3$ concentration compared to between that in large cities."

*20. L327: The OI effect in large cities – this is to be expected since emissions at surface are the dominant forcing and thus one would expect the OI signal to get swamped out more rapidly in locations with higher emissions.*

Response: This discussion here is meant to show the effectiveness and limitations of the OI method to adjust initial concentrations, and to provide quantitative results of the OI influence time in different cities in the LIS region. We added the discussion of using aloft measurements in the future when such data become available routinely.

*21. L330: please clarify if these durations of the difference imply a corresponding improvement also in model skill?*

Response: The different durations indicate the influence time of OI adjusted ICs, not necessarily the improvement in model skill, which is determined by both initial concentrations and other processes (chemical production and transport, etc). The improvement using the OI adjustment is similar in different subdomains (Table S3). We added this discussion to the revised manuscript (L410).

*22. L347-348: Consider qualifying this statement for the specific use in a forecast system. The NEI's are updated for this very reason - to capture changes in emissions. Perhaps they are not available in time for the forecast application and thus the need to project the NEI's to the forecast year. As written the sentence is somewhat open to misinterpretation – note that a specific year is attached to each NEI to indicate the period of its representativeness.*

Response: Thank you for this suggestion. It has been changed to

"This trend highlights the importance of updating the emissions to the model year, in order to reduce the bias in the emission inputs for model simulations, especially for time-sensitive applications such as air quality forecasting." (L434)

*23. L350: Please elaborate why the uniform and spatially varying emission adjustments result in similar predictions? Do the likely differences get averaged out in the aggregate comparisons?*

Response: There are several reasons causing the similar performance using the uniform and spatially varying emission adjustments. First, regional transport of air pollution results in dispersion of emitted $NO_x$ and its byproducts/reservoirs. The observations from satellite or ground monitors, based on which the emissions were adjusted, may not accurately capture the temporal evolution of the emission sources. A large geographical range may better reflect the overall changes of $NO_x$ emissions in the LIS region. Previous studies either use a coarse model resolution (e.g., 1 degree in Lamsal et al., 2011, or state-level adjustment in Tong et al., 2016). Second, the AQS sites in the city regions are usually located near high emission density areas. Similarly, satellite observations are weighted more toward urban plumes. The $O_3$ production in these places is $NO_x$ saturated, where the $O_3$ formation is less sensitive to changes in $NO_x$ emissions. As a result, the simulated concentrations using different methods were very close and the limited difference can also get averaged out when calculating the averaged statistical metrics. To illustrate the small difference, Table 4 and Table 5 have been added in the revised manuscript showing the metrics in

different subdomains. The tables and the discussion above have been added to the manuscript (L445).

Table 2: Statistical metrics of $O_3$ simulations after $NO_x$ emission adjustment in different sub-regions from August 26 to 31, 2018

| Domains\Stats | EmisAdj_avg | | | | EmisAdj_sub | | | |
|---|---|---|---|---|---|---|---|---|
| | CORR | RMSE | NMB | NME | CORR | RMSE | NMB | NME |
| NYC | 0.78 | 15.54 | -34% | 36% | 0.78 | 14.93 | -32% | 35% |
| PH | 0.78 | 15.29 | -30% | 35% | 0.78 | 15.38 | -31% | 35% |
| NHH | 0.85 | 13.24 | -25% | 31% | 0.85 | 13.24 | -25% | 31% |
| PP | 0.81 | 17.26 | -31% | 35% | 0.81 | 17.06 | -30% | 34% |
| OTHR | 0.84 | 12.24 | -24% | 29% | 0.84 | 12.17 | -24% | 29% |
| Average | 0.81 | 14.71 | -29% | 33% | 0.81 | 14.55 | -28% | 33% |

Reference
Lamsal, L. N., Martin, R. V., Padmanabhan, A., Van Donkelaar, A., Zhang, Q., Sioris, C. E., K. Chance, T. P. Kurosu, Newchurch, M. J. Application of satellite observations for timely updates to global anthropogenic NOx emission inventories. Geophysical Research Letters, 38(5), https://doi.org/10.1029/2010GL046476, 2011.
Tong, D., Pan, L., Chen, W., Lamsal, L., Lee, P., Tang, Y., Kim, H., Kondragunta, S. and Stajner, I.: Impact of the 2008 Global Recession on air quality over the United States: Implications for surface ozone levels from changes in NOx emissions, Geophys. Res. Lett., 43(17), 9280–9288, https://doi.org/10.1002/2016GL069885, 2016.

*24. L355-357: Please explain what "$O_3$ production is $NO_x$ saturated" implies - is ozone essentially titrated at the monitors examined? If so, shouldn't there be a low bias in the case where emissions were not adjusted? What LBCs did the emission adjustment runs use?*

Response: In the $NO_x$ saturated chemical regime, the ozone production efficiency is low, compared to that in a $NO_x$ sensitive chemical regime. In this case, $O_3$ level is not very sensitive to $NO_x$ change. Therefore, the difference in $O_3$ prediction between the two adjustment methods is small.

The default LBCs (same as in the Control run) were used here, since this section focused on the effect of a single adjustment (emission).

*25. L374-375: Please state what time average values these metrics are computed for and over what time-period.*

Response: This information has been added in the revised text (L487) and the caption of Figure 8:

"Figure 8 compares the predicted hourly $O_3$ and $NO_2$ concentrations against in-situ observations from August 26 to 31, 2018 in five subdomains and the overall domain with Taylor diagrams (Taylor, 2001)"

*26. L382: Please clarify what are the 3 adjusted runs? L370-374 indicate two adjusted runs? what does the reader associate runs #2,3,4 with?*

Response: We have clarified in the revised manuscript that there are three adjusted runs: BCON (dynamic BC), BO (dynamic BCs+OI) and BOE (dynamic BCs+OI+EmisAdj). We have changed the statement in L382 (L493 in revised manuscript):

"The three adjusted runs, namely BCON (#2), BO (#3) and BOE (#4) in diagrams, have …" to replace the "#2,3,4".

*27. L384-385: It was not clear to me how one could infer concentration levels and over/underestimation from the Taylor plots in Fig 8 – please elaborate.*

Response: Thank you. We added more information to help understand the diagram in L488: "In the Taylor diagram, the relative skill of each forecasting system to reproduce the $O_3$ and $NO_2$ variability is represented using three statistical metrics: correlation (R) with values on arc of the right angled sector, normalized standard deviation (SD) with values on y-axis, and centered root-mean-square difference (RMSD) with values on x-axis. The normalized SD is shown as the dashed line concentric circles while RMSD is shown as line concentric circles with the observation point acting as center (OBS on the x-axis). Their values higher (lower) than 1 indicate biased high (low) of the simulations. In general, the forecasting skill is measured by the distance to the OBS point on these diagrams."

*28. L390: Please reword this sentence - ICs impact the initial state, BCs do not.*

Response: This sentence has been rephrased (L506):

"Concurrent improvements of boundary conditions and initial concentrations allow a more realistic initial state and boundary conditions to demonstrate the effectiveness of the emission adjustment in improving $O_3$ forecasting (Fig. 9)."

*29. Fig 10 and associated discussion: Presumably, the NAQFC which used older emissions overestimated the $NO_x$ emissions relative to the BOE - why then does it consistently predict lower $NO_2$ relative to the BOE? Is it due to resolution or representativeness of emissions to the "forecast" year?*

Response: Yes, it is due to the spatial distribution of the emission. The NAQFC emission input is based on the 2011 National Emission Inventories (NEI), while the emission in the LIS system is updated to the forecast year (2018). The larger bias of $NO_2$ prediction is primarily attributed to the model resolution, with the higher resolution runs are more capable of describing the spatial distribution of emission sources.

*30. L459-460: Are the meteorological drivers also not different between NAQFC and BOE simulations? Could this also not influence the comparisons of the chemical constituents?*

Response: The BOE uses different meteorological fields from NAQFC, therefore the comparisons are certainly influenced by the difference in the meteorology. While a quantitative attribution to meteorology and emission differences is difficult to obtain, the magnitude of $O_3$ improvement from the base run to the BOE run is compared to that of reduced $O_3$ bias. We added the following statement to emphasize the limitation of this comparison (L616):

"Compared to NAQFC, BOE demonstrates enhanced prediction skills at all sites (Fig. 12). Note the comparisons may be attributed to the differences in meteorology, emission and other factors. Although it is difficult to attribute the improvement quantitatively to each factor, the magnitude of $O_3$ improvement from the base run to the BOE run is compared to that of the overall reduced $O_3$ bias, suggesting a significant contribution from these improvement techniques."

*31. L466: If the peaks are underestimated, then should it not be expected that the false alarms will also not increase? Not sure what aspect of the BOE the authors are attempting to highlight here? Also, the sample size (number of days, and sites) does not appear large enough to make a conclusive statement on FAR.*

Response: We have removed this statement.

*32. L470: Could the difference in the timing of the peak also result from differences in the meteorological fields?*

Response: Thank you for pointing out this. Research shows that emissions and meteorological fields are important in determining the magnitude and timing of high peaks of $O_3$ and $NO_x$, in particular wind direction is critical in determining the timing and location (Pan et al., 2017). Improving the meteorological fields will help improve the simulations of peak $O_3$ timing and concentrations (Bei et al., 2008). We also added this discussion in the text (L635):

"This may be in part attributed to the high resolution of the LIS3km system, which can better resolve meteorology and emission variations. As the emissions and meteorological fields play important role in determining the magnitude and timing of high peaks (Pan et al., 2017), emissions and meteorological data with 3 km resolution could improve the simulation of peak value and timing, especially over urban areas."

Reference

Bei, N., de Foy, B., Lei, W., Zavala, M., and Molina, L. T.: Using 3DVAR data assimilation system to improve ozone simulations in the Mexico City basin, Atmos. Chem. Phys., 8, 7353–7366, https://doi.org/10.5194/acp-8-7353-2008, 2008.

Pan, S., Choi, Y., Jeon, W., Roy, A., Westenbarger, D. A., and Kim, H. C.: Impact of high-resolution sea surface temperature, emission spikes and wind on simulated surface ozone in Houston, Texas during a high ozone episode, Atmos. Environ., 152, 362–376, https://doi.org/10.1016/j.atmosenv.2016.12.030, 2017.

**Responses to Reviewer #2:**
We thank the reviewer for the constructive comments, in particular the many valuable and detailed technical suggestions that have helped improve this manuscript. Please see below our point-to-point responses to these comments (*in Italic*).

Major Comments:
*1. Using static boundary conditions (BCs) is not a viable option for a small, polluted domain with well acknowledged transport impacts such as the northeastern US. Therefore, the results of applying static BCs are not interesting or useful. As expected, the model performs very poorly with static BCs. The discussion of this unviable option takes up unnecessary space and distracts the reader's attention up until the very end. The trivial conclusion that switching from static to dynamic BCs significantly improves ozone predictability undermines other (and in my view) more important improvements. I recommend removing the discussion of static boundary conditions from the main text. Including it as a supplement might be beneficial for novice readers.*

    Response: Thank you for your suggestion. We agree that the clean air BCs are not a viable option for the realistic air quality forecasting over the LIS region. In the final configuration that we used to simulate the high $O_3$ event, the NAQFC BCs were used. The static BC run was presented here for two purposes: First, it is used as the reference to assess the effectiveness of each adjustment method. Second, the difference between the Control run and the NAQFC BCs (BCON) run represents the contribution of regional transport of $O_3$ and its precursors to air quality over LIS. The results show that in-domain $O_3$ production remains relatively constant during peak hours (Figure 2d), but regional transport is key to allow the LIS system to reproduce high $O_3$ events. In addition, our results reveal that the influence of BCs vary by chemical species, more significant for long-lived species such as ozone but not so for nitrogen oxides, which are more sensitive to model resolution (spatially resolved emission and chemical transformation). We believe such results provide useful information to the literature, and are an integral part of this paper to make a full story. Following the reviewer' comment, we have shortened the discussion of static BCs and explained the role of the static BC simulation in the study design in Sect. 2.1 (L129):

    "The first group (Control run) applies no adjustment, using default profile as LBCs. It serves as the reference case to allow quantifying the effectiveness of each adjustment method."

*2. The rationale for using different variations of optimal interpolation (OI) could be presented better. The reader may not be familiar with the OI method and its strengths/weaknesses. Therefore, there should be a short discussion of the expectations with each alternative. The fact that the control case is no initial condition adjustment should be stated upfront otherwise the reader may think that the control is still the static boundary condition. The discussion of the performance with the inverse distance weighting option is insufficient or missing.*

    Response: Thank you. We have provided a general description of the OI method and the rationale with each alternative (OI_avg, OI_idw and OI_bias) in Sect. 2.3(b) L218: "Therefore, the region of influence is limited, and the adjusted fields may be discrete in spatial distribution." L221: "With this interpolation, the effect of OI will be not limited near the observational sites and most of the grid cells in the domain can be adjusted comparing to the OI_avg.", and L225: "Unlike the OI_idw which just applied the spatial interpolation to extend the OI effect, in this method the observation cells are distributed to the whole domain grids based on the spatial patterns provided by model so that it is able to better reflect the realistic fields."

We have made it clear in the revised manuscript that the control case uses the static BCs (Section 3.2):

"Here we compare the results using various OI methods with the simulations without any BC adjustment (same as the Control run)."

In addition, we modified Table 1 to clarify the model settings for each run. Finally, we added more description of the performance of each method (Lines 372-378):

"The predicted $O_3$ field with the OI_avg method shows a distribution with localized high value areas near the observation sites. As for the other two OI methods, the distribution using OI_bias has similar patterns with that of OI_idw while the concentrations over the high $O_3$ area are further elevated. Biases between observed and predicted concentrations are reduced in most of the areas. The statistical metrics calculated from hourly simulated and observed data from August 26 to 31, 2018 were reported in Table 2. The RMSEs for $O_3$ are reduced from 14.97 ppbv to 13.72 ppbv in the OI_bias run, to 13.79 ppbv in the OI_idw run and to 14.30 in the OI_avg run."

*3. I disagree with the choice of the domain-average emission adjustment based on performance. The $NO_x$ emission differences between the base year (2011) and current year (2018) are so different for the four subdomains that averaging them cannot be justified. The subdomain emission adjustment is clearly the right choice because it provides the model with the right information. The better performance with the domain average emission adjustment here is a typical case of getting the "right" answer for the wrong reason. I recommend a more detailed, site specific analysis of performance with these two adjustment methods. I expect at least the $NO_2$ performance of the subdomain adjustment to be better at the sites in and around those subdomains.*

Response: We agree that the subdomain adjustment makes more sense and has initially hypothesized that the results using subdomain emission adjustments would be better. To answer this question, we have conducted the model simulations with both full domain adjustment and subdomain adjustment (EmisAdj_sub) during the study period. We found that the performance from the model runs with different adjusting methods was quite similar. So we didn't put the results of EmisAdj_sub in the original manuscript. There are several reasons causing the similar performance using the uniform and spatially varying emission adjustments. First, regional transport of air pollution results in dispersion of emitted $NO_x$ and its byproducts/reservoirs. The observations from satellite or ground monitors, based on which the emissions were adjusted, may not accurately capture the temporal evolution of the emission sources. A large geographical range may better reflect the overall changes of $NO_x$ emissions in the LIS region. Previous studies either use a coarse model resolution (e.g., 1 degree in Lamsal et al., 2011, or state-level adjustment in Tong et al., 2016). Second, the AQS sites in the city regions are usually located near high emission density areas. Similarly, satellite observations are weighted more toward urban areas where $O_3$ production is $NO_x$ saturated and less sensitive to changes in $NO_x$ emissions. As a result, the simulated concentrations using different methods were very close and the limited difference can also get averaged out when calculating the averaged statistical metrics.

Following the reviewers' suggestion, Table 4 and Table 5 have been added in the revised manuscript showing the metrics in different subdomains. The tables and the discussion above have been added to the manuscript (L446-454):

"This demonstrates that emission adjustment alone results in limited improvement of $O_3$ prediction, due in part to the fact that the $O_3$ production in this region is $NO_x$ saturated (VOC limited) in urban areas where most AQS monitors are deployed, so the $O_3$ level is less sensitive to the change in $NO_x$ emissions. Similarly, satellite observations are weighted more toward urban

plumes. In addition, regional transport of air pollution results in dispersion of emitted $NO_x$ and its byproducts/reservoirs. The observations from satellite or ground monitors, based on which the emissions were adjusted, may not accurately capture the temporal evolution of the emission sources. A large geographical range may better reflect the overall changes of $NO_x$ emissions in the LIS region. Previous studies either use a coarse model resolution (e.g., 1 degree in Lamsal et al., 2011, or state-level adjustment in Tong et al., 2016). As a result, the simulated concentrations using different methods were very close and the limited difference can also get averaged out when calculating the averaged statistical metrics."

**Table 4: Statistical metrics of O$_3$ simulations after NO$_x$ emission adjustment in different sub-regions from August 26 to 31, 2018**

| Domains/Stats | EmisAdj_avg | | | | EmisAdj_sub | | | |
|---|---|---|---|---|---|---|---|---|
| | CORR | RMSE | NMB | NME | CORR | RMSE | NMB | NME |
| NYC | 0.78 | 15.54 | -34% | 36% | 0.78 | 14.93 | -32% | 35% |
| PH | 0.78 | 15.29 | -30% | 35% | 0.78 | 15.38 | -31% | 35% |
| NHH | 0.85 | 13.24 | -25% | 31% | 0.85 | 13.24 | -25% | 31% |
| PP | 0.81 | 17.26 | -31% | 35% | 0.81 | 17.06 | -30% | 34% |
| OTHR | 0.84 | 12.24 | -24% | 29% | 0.84 | 12.17 | -24% | 29% |
| Average | 0.81 | 14.71 | -29% | 33% | 0.81 | 14.55 | -28% | 33% |

**Table 5: Same with Table 4 but for NO$_2$**

| Domains/Stats | EmisAdj_avg | | | | EmisAdj_sub | | | |
|---|---|---|---|---|---|---|---|---|
| | CORR | RMSE | NMB | NME | CORR | RMSE | NMB | NME |
| NYC | 0.82 | 4.23 | -22% | 27% | 0.82 | 4.77 | -29% | 31% |
| PH | 0.79 | 5.69 | -36% | 41% | 0.79 | 5.53 | -33% | 40% |
| NHH | 0.49 | 7.69 | -44% | 49% | 0.49 | 7.53 | -41% | 48% |
| PP | 0.67 | 2.92 | -18% | 35% | 0.67 | 2.95 | -21% | 36% |
| OTHR | 0.69 | 2.56 | -33% | 39% | 0.69 | 2.54 | -32% | 39% |
| Average | 0.69 | 4.62 | -31% | 38% | 0.69 | 4.67 | -31% | 39% |

Reference

Lamsal, L. N., Martin, R. V., Padmanabhan, A., Van Donkelaar, A., Zhang, Q., Sioris, C. E., K. Chance, T. P. Kurosu, Newchurch, M. J. Application of satellite observations for timely updates to global anthropogenic NOx emission inventories. Geophysical Research Letters, 38(5), https://doi.org/10.1029/2010GL046476, 2011.

Tong, D., Pan, L., Chen, W., Lamsal, L., Lee, P., Tang, Y., Kim, H., Kondragunta, S. and Stajner, I.: Impact of the 2008 Global Recession on air quality over the United States: Implications for surface ozone levels from changes in NOx emissions, Geophys. Res. Lett., 43(17), 9280–9288, https://doi.org/10.1002/2016GL069885, 2016.

*4. Section 4 should include the comparisons of NO$_2$ column and O$_3$ profile measurements during LISTOS with the model using the subdomain emission adjustment.*

Response: Thank you. Following the reviewer's suggestion, we have added the comparisons of NO$_2$ column and O$_3$ profile measurements using the subdomain emission adjustment.

In Sect. 3.4, we added the comparisons between results of domain average adjustment (EmisAdj_avg) and subdomain adjustment (EmisAdj_sub). The results show that the model performance between the two runs is similar. (L483: First, we compare two BOE simulations, one with the EmisAdj_avg emission adjustment and the other with EmisAdj_sub. The statistical metrics of BOE with EmisAdj_avg and BOE with EmisAdj_sub (Table S4, S5) are quite similar in each sub region and also have the same correlations. On average, the RMSEs of BOE (EmisAdj_avg) is slightly smaller. Therefore, in the subsequent evaluation we take BOE (EmisAdj_avg) to compare against surface and other observations.)

In Sect. 4, we added the comparison of NO$_2$ column and O$_3$ profile simulated by combined adjustment using EmisAdj_sub (L595):

"In addition, the NO$_2$ VCD from simulation with combined adjustments using EmisAdj_sub method for emission refresh shows a similar spatial pattern with that of BOE (Fig. S3) while its VCD level over the NYC area is lower, making it underestimates the hotspot but much closer to the VCD over the rest of the areas."), L601 (and unlike the results of surface NO$_2$, the NO$_2$ VCD using EmisAdj_sub has lower NMB (33%) and NME (57%) compared to that using EmisAdj_avg (40% and 61%) while their correlation is still the same (0.74).) and L668 ("The model performance of O$_3$ surface concentrations and vertical distribution using AFs from EmisAdj_sub is very close to those of using the AFs from EmisAdj_avg in the BOE case (Fig S4, Table S7)")

All results of these comparisons (Figures and Tables) have been included in the Supplementary Information.

*Specific Comments:*

*1. L31: "derived from NOAA National Air Quality Forecast Capability (NAQFC)." What is the forecasting system used here? You should give it a name.*

Response: NAQFC is the name of the NOAA operational forecasting system. We have added the information here.

*2. L73: Tong and Tang, 2018. There are certainly older references to cite here.*

Response: Thank you. Two older references were added:

Eder, B., Kang, D., Rao, S. T., Mathur, R., Yu, S. C., Otte, T., Schere, K., Wayland, R., Jackson, S., Davidson, P., and McQueen, J.: A demonstration of the use of national air quality forecast guidance for developing local air quality index forecasts, B. Am. Meteorol. Soc., 91, 313–326, doi:10.1175/2009BAMS2734.1, 2010.

Oliveri Conti, G., Heibati, B., Kloog, I., Fiore, M., Ferrante, M. A review of Air QModels and their applications for forecasting the air pollution health outcomes.Environ. Sci. Pollut. Res.http://dx.doi.org/10.1007/s11356-016-8180-1, 2017.

*3. L113: (NAQFC). You need a reference here.*
Response: A reference has been added:

Davidson, P., Schere, K., Draxler, R., Kondragunta, S., Wayland, R. A., Meagher, J. F., and Mathur, R.: Toward a US National Air Quality Forecast Capability: Current and Planned Capabilities, in: Air Pollution Modeling and Its Application XIX, edited by: Borrego, C. and Miranda, A., pp. 226–234, Springer, Dordrecht, The Netherlands, 2008.

*4. L119: NAQFC. State the resolution (12 km)*
Response: Changed to "… NOAA NAQFC with a horizontal resolution of 12 km were applied …"

*5. L182: $X_b$. Undefined*
Response: We have added the definition here:
"where $x^a$ and $x^b$ are the analyzed and background fields, respectively."

*6. L190: "Experiments were also performed with two different interpolation methods." Are these the same OI methods mentioned above. This paragraph is confusing.*
Response: Thank you for pointing out this. Yes, these are the same OI method mentioned above and the interpolation methods here indicate the method is used for preparing observational data for OI. We remove the first sentence in this paragraph as it was repetitive with L190, and edit this as "Besides this method, experiments were also performed with two different interpolation methods for preparing the observational data."

*7. L213: " $f_S$ is set to 1 and $f_G$ to 100 to avoid dominance by either data source." Sounds like surface data dominance.*
As the monthly OMI data is calculated from the daily files, its temporal resolution is relatively lower. In comparison, the ground-based (AQS) $NO_2$ is measured more frequently (hourly) and can better reflect the local emission situation. Therefore, the AQS data is considered more reliable for calculating the emission changes. Additionally, in the formula, $N_S$ (number of satellite data) and $N_G$ (number of AQS data) depend on the number of grid points in the OMI file and number of observational sites, respectively. So $N_S$ is much larger than $N_G$. We set $f_G$ to 100 to avoid the dominance by OMI data.

*8. L216: AFs from May to September. Why do you call it "rapid" refresh?*
Response: The rapid refresh means the emission input can be adjusted quickly using the observations, compared to the typical time lag of several years in the model emission input files from national emission inventories. This approach has the potential to update emission as soon as the observations are made available (within a day from the forecast time).

*9. L225: obtained from the AQS network. Repetition*
Response: We have removed "obtained from the AQS network and"

*10. L236: Ozone Monitoring Instrument. Defined earlier*
Response: We have removed the "Ozone Monitoring Instrument" here.

*11. L237: aboard the Aura satellite. Repetition*
Response: Removed. Thank you.

*12. L242: Multiyear OMI NO₂ data were further aggregated to calculate state-level emission adjustment factors using a mass conservation approach (Tong et al., 2015). This does not say much. Why mass conservation is mentioned here?*
Response: Unlike the concentrations, the fluxes and emissions are related to the area and time. So it is necessary to keep the total emission amount constant during the data processing. The "using a mass conservation approach" indicates the method will keep the processed $NO_2$ emission consistent with the original one.

*13. L245: GeoCAPE Airborne Simulator. Previously defined.*
Response: Removed.

*14. L252: Langley Mobile Ozone Lidar. Previously defined.*
Response: Removed.

*15. L253: LMOL is part of a NASA-sponsored ozone lidar network called the tropospheric ozone lidar network (TOLNet; Sullivan et al., 2019). Why is this relevant?*
Response: Here we provide a short introduction of the lidar data and relevant reference for the readers who may be interested in learning more about LMOL.

*16. L262: "As a reference." Why? NAQFC has 12-km resolution.*
Response: Here we introduce the NAQFC product as a reference/benchmark to compare and evaluate the new LIS 3 km system, aiming to evaluate and analyze the performance of the new high-resolution forecasting.

*17. L265: "This suggests the default profiles provided by CMAQ represent a clean environment, such as marine air layer, and are not suitable for areas with active emissions and tropospheric O₃ production." Well, obviously. There is nothing new or original here. Nobody uses default BCs for their region.*
Response: We moved this statement and further discussed the influence of dynamic BC in Sect. 3.1 (Major comment 1).

*18. L275: "The performance of the two high-resolution simulations was next compared to that by the NAQFC." Again, why? You should do this when you put in all the upgrades.*
Response: The comparison here is for the simulations with each adjustment method, the single method adjusted results in Sect. 3.1~3.3 can compare with the NAQFC results and present the effectiveness of each adjustment method. It also showed the differences of high-resolution simulations with NAQFC on spatial patterns and metrics to the reader at first sight. In Sect. 3.4 and Sect. 4, we did further inter-comparisons for the combined adjustment results.

*19. L286. Are there any tables or figures to support these statements?*

Response: Thank you. We have added Table S2. in the Supplementary Information to support these statements.

*20. L294. I recommend removing the static boundary condition. This section gives the impression that static boundary condition is better than coarse resolution. This may be the case for $NO_2$ but overall, when $O_3$ and $NO_2$ are considered together, static boundary conditions are worse.*

Response: Thank you. We have responded to this comment earlier (see Response to Major Comment #1).

*21. L299. Is the control run still the static BC run? Or, is it the original OI?*

Response: Yes, the Control run uses static BCs, not with the OI adjustment. We have explicitly stated it in the revised manuscript (L356) and further clarified the settings for each run in the revised Table 1.

*22. L302: Fig. 4a. Do you want us to compare 4a and 4b?*

Response: Thank you for pointing this out. It has been revised:

"… regions within five model grid cells in each direction of the observations compared to the Control run (Fig. 4a, b)".

*23. L303: (Fig. 4b, c). Which panels do you want us to compare?*

Response: Thank you. It has been changed to 4c,d: "The $O_3$ fields adjusted by OI_idw (Fig. 4c) and OI_bias (Fig. 4d) show similar horizontal distributions."

*24. L308: "that." than*

Response: We have revised this sentence to "The adjusted $O_3$ fields show different patterns compared to that in the Control run with no IC adjustment."

*25. L308: "the Control run with no IC adjustment." You should define the control run earlier.*

Response: The Control run here is the same as in the previous sections. We have added the definition of each run Table1 of the revised manuscript.

*26. L310: "The RMSEs for $O_3$ are reduced from 14.97 ppbv to 13.72 ppbv in the OI_bias run and to 14.30 in the OI_avg run." What about idw?*

Response: Here we added the description for metrics of OI_idw: "the RMSEs for $O_3$ are reduced from 14.97 ppbv to 13.72 ppbv in the OI_bias run, to 13.79 ppbv in the OI_idw run and to 14.30 in the OI_avg run." (L378)

*27. L314: "Generally, the improvement of the simulated results due to OI data assimilation over the study domain is smaller than that from the dynamic BCs for this particular domain." You are comparing apples with oranges.*

Response: All LBCs and OI runs are compared to the same Control run to assess the effectiveness of different adjustment methods.

*28. L316: "may yield different relative changes between BCs and ICs adjustments." Of course.*

Response: Thank you. We have removed this sentence.

*29. L319: "the duration of OI influence." Perhaps you should remind that you run each day with new ICs.*

Response: Thank you. We added the information in Sect. 2.3 (L203):

"In the CMAQ model, the restart file, called CGRID, is daily generated during the simulation and acts as ICs for the next day. To constrain the biases in ICs, the concentrations of ozone, $NO_2$ and NO in the restart file were adjusted via the OI method, which is applied every 24 hours at 0:00 Coordinated Universal Time (UTC)."

Again in L397:

"The ICs for each day were adjusted by OI using real time observations."

*30. L340. Delete "emissions".*

Response: Removed.

*31. L341: Add "in ground concentrations"*

Response: Thank you. Added "in ground concentrations" to the text.

*32. L343: "The average adjustment factor (AF)". AF already defined, just refer back to Equation 2.*

Response: Revised this into "The average AF for …".

*33. L347: "input" should be inputs.*

Response: Revised this into "inputs". Thank you.

*34. L352: delete "as"*

Response: Removed "as".

*35. L362: "Considering both $O_3$ and $NO_2$ performance, the domain-average approach (EmisAdj_avg) is selected for subsequent multi-adjustment simulations." This is counterintuitive. Theoretically, the subdomain approach should be better because it gets you closer to reality, does it not? There is much to be analyzed here. Just because performance is better with the domain-average approach does not mean that is how one should proceed. An analysis of how individual monitors behave is recommended. You are getting a better result for the wrong reason. You should proceed with the better approach.*

Response: We analyzed the reasons why the results from domain-average and subdomain AFs are so similar. We also provided the statistical metrics in each subregion and added the simulation results using the subdomain AF in the relevant sections. See details in the responses to Major Comments #3 and #4,

*36. L377: "centered root-mean-square difference (RMSD)." Explain how to read this on the diagram.*

Response: We added the instruction to understand RMSD on the diagram (L489):

"In the Taylor diagram, the relative skill of each forecasting system to reproduce the $O_3$ and $NO_2$ variability is represented using three statistical metrics: correlation (R) with values on arc of the right angled sector, normalized standard deviation (SD) with values on y-axis, and centered root-mean-square difference (RMSD) with values on x-axis. The normalized SD is shown as the dashed line concentric circles while RMSD is shown as line concentric circles with the observation point

acting as center (OBS on the x-axis). Their values higher (lower) than 1 indicate biased high (low) of the simulations. In general, the forecasting skill is measured by the distance to the OBS point on these diagrams."

*37. L381: Should be Fig. 8e.*
Response: Thank you so much. We have changed to Fig. 8e

*38. L382: "The three adjusted runs (#2, 3, 4)". Is this the same as BOE?*
Response: These runs are three simulations with different adjustment settings: dynamic BCs alone (BCON), dynamic BCs and OI (BO), and all three methods together (BOE). We revised the text (L499):
"The three adjusted runs, namely BCON (#2), BO (#3) and BOE (#4) run in diagrams, have well reproduced surface $O_3$ concentrations over the NYC region."

*39. L396: "All of the high-resolution simulations, including the Control run, perform better than the NAQFC run (Fig. 9)". For NO₂, right?*
Response: Yes, this sentence has been revised:
"All of the high-resolution simulations, including the Control run, perform better for $NO_2$ prediction than the NAQFC run ..."

*40. L398: "Since the boundary conditions used by the high-resolution CMAQ runs are derived from NAQFC, the large negative NO₂ bias from NAQFC also contributes to the overall bias to the high-resolution runs." This contradicts what you said earlier about NO₂ when you were comparing static versus dynamic BCs. You said BCs did not matter for NO₂.*
Response: The $NO_x$ is primarily influenced by local emissions because of its short lifetime, the transported source is not as significant as that of $O_3$, although the effect is not zero. We have removed this statement.

*41. L403: "Such a contrast suggests either an underestimate of emission sources in Connecticut, or flawed model chemistry and transport, or a combination of both." What were the results for subdomain emissions corrections?*
Response: The results for subdomain emissions are analyzed in L446 (as discussed in the response to General Comment #4) of revised manuscript and the metrics are put in Table S4 and Table S5.

*42. L408: "(August 28-29, 2018)". What is different from the week of August 25-31 stimulated before?*
Response: It's the same simulation covered in this study. The $O_3$ concentrations were very high (exceeding the $O_3$ air quality standards or NAAQS) on August 28-29, 2018 and there are LISTOS observations during that time. So here we focus on these two days which have more observational dataset (aircraft and lidar). We change this to "During the high $O_3$ pollution days (August 28-29, 2018) in this episode".

*43. L446: "(Fig. 11e, 11i)". Labels are missing hence I am unable to follow this discussion.*
Response: It has been corrected now. See the revised Figure 11 and caption.

[Figure]

44. L466: "false alarms". Do you mean a predicted ozone exceedance that does not realize?
Response: Yes, "false alarms" are used to refer to a predicted exceedance that is not observed. Here the overestimation on August 29, 2018 was not large enough to trigger a false alarm. We have removed this statement to avoid confusion.

45. L481: "O$_3$ concentrations in the free troposphere are more controlled by regional O$_3$ production and transport than in the PBL." Another reason why static BCs should not be used.
Response: Yes, and the effect of BCs on local O$_3$ also depends on the simulated seasons. We had a discussion about this in the previous section.

46. L498: "although the system needs to be further refined to reduce bias." The subdomain emission refresh should be explored more in this simulation.
Response: The results of BOE using the subdomain emission refresh have been added.

47. L507: "reproduce" to "simulate"
Response: Changed.

48. L520: "One possible direction to explore is to use atmospheric observations to constrain emissions through coupled data assimilation approaches in future efforts." Unclear
Response: We have revised this sentence to add more information. Since we only applied a simple emission adjustment approach here, there are other ways to constrain emissions. This sentence has been changed to (Lines 699-702):
"One possible direction to explore is to apply other methods to constrain emissions that use both variational (e.g. Elbern et al., 2007; Vira and Sofiev, 2012) and ensemble-based (e.g. Miyazaki et al., 2012, 2017) solutions to analyze the 3D chemical tracers as well as their respective precursor emissions simultaneously."
Reference:

Elbern, H., Strunk, A., Schmidt, H., and Talagrand, O.: Emission rate and chemical state estimation by 4-dimensional variational inversion, Atmos. Chem. Phys., 7, 3749–3769, https://doi.org/10.5194/acp-7-3749-2007, 2007.

Miyazaki, K., Eskes, H. J., and Sudo, K.: Global $NO_x$ emission estimates derived from an assimilation of OMI tropospheric $NO_2$ columns, Atmos. Chem. Phys., 12, 2263–2288, https://doi.org/10.5194/acp-12-2263-2012, 2012.

Miyazaki, K., Eskes, H., Sudo, K., Boersma, K. F., Bowman, K., and Kanaya, Y.: Decadal changes in global surface $NO_x$ emissions from multi-constituent satellite data assimilation, Atmos. Chem. Phys., 17, 807–837, https://doi.org/10.5194/acp-17-807-2017, 2017.

Vira, J., & Sofiev, M. On variational data assimilation for estimating the model initial conditions and emission fluxes for short–term forecasting of $SO_x$ concentrations. Atmospheric Environment, 46, 318–328. https://doi.org/10.1016/j.atmosenv.2011.09.066, 2012

49. L526: GCAS. Are you redefining all the acronyms in the summary?
 Response: We removed the repeat definition.

*50. L531: Data Availability. What about the other data?*
Response: In the revised manuscript we have provide the sources of the other data used in this study:
    "CMAQ and SMOKE source code is available on the Community Modeling and Analysis System (CMAS) Center of University of North Carolina, Chapel Hill: https://www.cmascenter.org/ (last access: July 31, 2021). WRF is an open-source community model. The source code is available at http://www2.mmm.ucar.edu/wrf/users/download/get_source.html (last access: July 31, 2021). The AirNOW hourly data of $O_3$ and $NO_x$ is available at https://files.airnowtech.org/?prefix=airnow/ (last access: May 2021) and the hourly $NO_x$ data from US EPA Air Quality System (AQS) surface network is available at https://aqs.epa.gov/aqsweb/airdata/download_files.html (last access: May 2021). The GCAS $NO_2$ vertical column density and the LMOL $O_3$ vertical profile data are available at https://www-air.larc.nasa.gov/missions/listos/index.html (last access: May 2021). The monthly product of $NO_2$ vertical column density from OMI is available at https://avdc.gsfc.nasa.gov/pub/data/satellite/Aura/OMI (last access: July 31, 2021)"

*51. Table1. Is the control the same in all the figures? The control in Figure 2 is much different than the control in Figures 4 and 5.*
Response: Yes, this Control run is the same in all figures. In Figure 2, the spatial distribution is daily concentration, the values are averaged, while Figures 4 and 5 show the hourly distributions.

*52. Figure2. The label on the panel is "default".*
Response: We have changed it to the Control run, which uses the default BC profile in CMAQ.

*53. Figure 6. Better call it OI*
Response: Thank you for your suggestions. Revised this to "OI adjusted initial concentrations".

*54. Figure 8. Change standardized to standard on the y axis.*
Response: Thank you. It has been changed in the revised Figure 8.

[Figure]

*57. Figure 9. The magnitude cannot be % in a and b.*
Response: This was a typo and has been corrected. Thank you!

[Figure]

58. *Figure 10. Label these sites on Figure 1.*

Response: Thank you. The labels have now been added to Figure 1:

[Figure]

Figure 1: Study area over the Long Island Sound and the surrounding region. Red boxes depict four subdomains: New York City (NYC), Philadelphia (PH), New Haven-Hartford region (NHH), and Providence-Pawtucket region (PP). Black circles indicate the locations of EPA air quality system (AQS) ground monitors, the brown triangle indicates the TOLNet $O_3$ site located in Westport, CT, and the blue lines present an example flight path conducted by the NASA B200 aircraft on August 28~29, 2018. Letters a–j indicate the monitoring sites: a) Flax Pond, b) Queens College, c) New Haven, d) Westport, e) Colliers Mills, f) Riverhead, g) Greenwich, h) Madison-Beach Road, i) Middletown-CVH-Shed and Stratford.

---

## Author Response (AR2)

**Responses to Reviewer #1:**

We appreciate the reviewer for taking time to review the revised manuscript and provide new comments. Please see below our point-to-point responses to these comments (*in Italic*).

*1) L202-204: Please indicate at what model levels (altitudes) the ICs for $O_3$, NO, $NO_2$ were adjusted by the OI method – I believe it is only at the surface, but for completeness it should be explicitly stated.*

Response: The ICs were adjusted for all layers within the PBL. We have explicitly stated it in Line 211:

"Next, as there is no information of vertical background profile in this method, the ratio between $x^a$ and $x^b$ at each surface layer grid point was used to scale the concentrations **for all layers within the PBL**, following Tang et al. (2015; 2017)."

*2) The discussion related to the default BCs as a "control run", its representativeness for the conditions being simulated, and inferences drawn relative to the case with "dynamic" BCs still raise some questions. While it is okay to convey it as a control run, I feel some of the inferences drawn and the suggested importance of the use of the default BCs need to more carefully conveyed. If the intent of the "control" run is to infer the pollution burden solely associated with emissions within the LIS domain, wouldn't a case with no inflow (i.e., zero boundary conditions) be more appropriate? The static default LBCs appear to be so unrepresentative for the small LIS domain with surrounding regions of relatively high emissions, that anything with even slightly better representativeness would yield improved performance for the aggregate statistics by reducing the systematic biases induced by the default LBCs. I am thus a bit wary of some of the broad conclusions that are drawn on the suitability of one over the other for different seasons.*

Response: Thank you. Following the suggestion, we have tuned down the BCs discussion and emphasized on the limitation in the conclusion drawn from this case study. See detailed changes below:

L326:

"For instance, the $O_3$ concentration over the Long Island Sound is lower than its surrounding areas during this episode."

L329:

"This indicates the high-resolution simulation can better reproduce the pollutant variability over this coastal urban area during this study period."

*3) For instance, the discussion starting on L304 states that the "influence of BCs is less important during the cold season" – I would exercise some caution in making such broad sweeping statements based on the limited domain extent and conditions examined here. In fact, there are many studies documenting the importance of BC specification during times when "local" pollution production is slow (as in cool seasons). At the very least, the statement should be qualified to convey the specific and limited conditions it is based on. By the same token, I do not believe the importance of specifying space and time varying BC, especially for small geographic domains as the LIS region should be diminished just because deriving them from larger scale simulations is considered computationally burdensome for real-time applications (as suggested in the author response).*

Response: Thank you. We made this statement (influence of BCs is less important during the cold season) based on the observation that there is a smaller difference between the upwind concentrations and the background concentrations used in the default BCs, compared to that during a hot season when the upwind photochemical production is more active (Fig. S2a, d). We agree that there could be cases in which long-range transport becomes even more important during winter. Therefore, we have modified the statement and discussed the aforementioned limitation (L308):

"In comparison, the influence of BCs is less important during the cold season. **There is a smaller difference between upwind concentrations and the background concentrations used in the default BCs, compared to that during a hot season when the upwind photochemical production is active,** resulting in better agreement between the prediction and observations (Fig. S2a, d). **Note that other studies have shown the influence of BCs becomes more prominent during the cold season when "local" pollution production is slow (e.g. Fiore et al., 2009). The magnitude of the actual influence is determined by several factors, such as the emission density, photochemical production and sink, and spatial range and gradients of the concerned species.**"

We agree that the importance of deriving the dynamic BCs should not be diminished, which is the motivation to test the NAQFC BCs, which are provided by NOAA's operational prediction and readily available for regional modelers to use.

Reference:
Fiore, A. M., Dentener, F. J., Wild, O., Cuvelier, C., Schultz, M. G., Hess, P., Textor, C., Schulz, M., Doherty, R. M. and Horowitz, L. W.: Multimodel estimates of intercontinental source-receptor relationships for ozone pollution, J. Geophys. Res. Atmos., 114(D4), 2009.

4) I don't believe the conclusion on L311 that influence of dynamic BCs is "more significant during the high pollution time" is robust. Dynamic BCs could also be influential in conditions when local production may not be high. The dynamic BCs help better represent the changes in baseline pollution at a location based on changes in large scale forcing and thus should be important at all times. As stated, I think the statement can be easily misconstrued and should either be qualified for specific conditions, or further substantiated, or deleted. It appears that for the cases examined in this study the baseline levels in the LIS were enhanced by regional transport.
Response: We removed this statement.

5) L323: I think this sentence is somewhat trivial in that it should be expected that a finer resolution simulation will show "more detailed distributions" than a coarser resolution simulation – perhaps it would useful to point out specific features of interest that are better resolved and point the reader to the appropriate illustration supporting that. How does one assess if the more detailed distribution is better?
Response: We added in the revised manuscript the specific features of the spatial differences between the two simulations (L326):

"For instance, the $O_3$ concentration over the Long Island Sound is lower than its surrounding areas during this episode, which is better resolved by the 3 km simulation than the 12 km NAQFC (Fig.

2a-c). The O₃ distribution along the coastal area, such as the coasts of Connecticut and Rhode Island, also agrees better with the observations than the 12 km NAQFC prediction."

Response: We have specified that this is the case during the study period, as shown in Figure 2.

"the O₃ concentration over the Long Island Sound is lower than its surrounding areas **during this episode**"

Response: Thank you. We have added the following explanation in the revised manuscript (L328):

"The O₃ distribution along the coastal area, such as the coasts of Connecticut and Rhode Island, also agrees better with the observations than the 12 km NAQFC prediction."

Response: Thank you. Following this suggestion, we have tuned down this statement (L329):

"This indicates the high-resolution simulation can better reproduce the pollutant variability over this coastal urban area **during this study period**."

Response: Revised to "2014 NEI". Thank you.

Response: The statement "the grid resolution solely resulted in the NO₂ spatial pattern differences between these runs and the NAQFC" cannot be found in L581 or anywhere in the revised manuscript.

Response: We have added the illustration (Figure 12) into this sentence (L644):

"As the emission and meteorological inputs play an important role in determining the magnitude and timing of high peaks (Pan et al., 2017), high resolution emissions and meteorological data

contributed to the improved simulation of peak **O$_3$** value and **its** timing, especially over urban areas **(Fig. 12).**"

12) In response to my query on whether emissions of species other than NO$_x$ were "refreshed", the authors response states that "regional O$_3$ production is more controlled by NO$_x$ than VOC" – while this may be true for regional O$_3$ distributions, does it also hold for the NY metro area and the LIS region. I do acknowledge, that changing emissions over the past several decades has changed NO$_x$ and VOC sensitive regimes in the area, and would think that capturing both changes in NO$_x$ and VOC emissions would be important. It is also important to note that several recent studies suggest perhaps higher VOC emissions from volatile consumer products than represented in the inventories. Thus, it seems that updates to VOC emissions may also influence O$_3$ simulations in some urban areas. It may thus be worthwhile to at least acknowledge the need for updates similar to NO$_x$ for other precursor species, while also acknowledging the challenges in doing so.

Response: Thank you. We have acknowledged the need for VOCs updates simiilar to NOx (L710):

"In addition, the importance of volatile consumer product VOCs has been identified in recent studies (McDonald et al., 2018), suggesting that updating other species than NO$_x$ is also necessary. This may be challenging through a similar approach, due to limited measurements of VOCs from both ground and space instruments."

13) L402: the sentence "In large metropolitan areas, OI adjustments result in spikes in large metropolitan areas indicate the model errors at the time..." is awkwardly phrased and needs to be rewritten.

Response: Thank you. We have rephrased this sentence (L404):

"In large metropolitan areas, OI adjustments result in spikes  that indicate  larger model errors at the time of OI adjustment, with the mean errors  up to 14 ppbv in surface hourly O$_3$ concentrations over NYC and 16 ppbv over Philadelphia, respectively."

14) The additional tables 4 and 5 only re-enforce that the two adjustment techniques did not alter performance much, not necessarily explain likely reasons. I am not sure the various possible reasons outlined in the response, coherently suggest the reason either. If O$_3$ production is low in "NO$_x$ saturated" regions, why compare O$_3$ there rather than in downwind regions where the production is enhanced? Is the suggestion that in downwind regions the urban emissions in the plume are "well-mixed" such that the spatial adjustments in emissions in the urban core have little effect? If so, it may be worthwhile speculating so.

Response: Tables 4 and 5 provided here cover regions with both high (four urban areas, NYC, PH, NHH and PP) and low (OTHR) emission density. Therefore, these tables show the influence of NO$_x$ emission adjustments over both urban and well-mixed downwind regions.

**Responses to the Comments by Reviewer #2:**
We want to sincerely thank the reviewer for the helpful comments in the second round. We provide below the point-to-point responses to these comments, and corresponding changes have been made in the revised manuscript.

1. Line 189: "BCs that can be used" (insert "can")
Response: The word has been inserted:
 "BCs that **can** be used to drive"

2. Line 228: "provided by the model" (insert "the")
Response: Inserted.

3. Lines 401-402: delete the second "in large metropolitan areas" and insert "that" after spikes
Response: This sentence has been modified as follows (L403),

"In large metropolitan areas, OI adjustments result in spikes  **that** indicate  **larger** model errors at the time of OI adjustment , with the mean errors  up to 14 ppbv in surface hourly $O_3$ concentrations over NYC and 16 ppbv over Philadelphia, respectively. "

4. Line 449: Elaborate what you mean by "Similarly, satellite observations are weighted more toward urban plumes."
Response: Changed to "Similarly, the retrievals of satellite observations are also more sensitive to urban plumes."

5. Line 486-487: "On average, the RMSEs of BOE (EmisAdj_avg) is slightly smaller." This is not true for ozone (Table 4),
Response: You are right that it is not true for ozone in Table 4. However, Table 4 describes the performance for the designed EmisAdj studies that are different from the final configuration (with BCON, OI and Emission adjustment or so called BOE). The referred statement was intended for the BOE case in Table 4&5, not in Table 4. The results are slightly different once all adjustment methods are used together from that when a single method was applied. We have clarified it in the revised manuscript.

"On average, the RMSE of the combined BOE setup is slightly smaller using the EmisAdj_avg method than that using the EmisAdj_sub method during the study period (Table S4, S5)**, which is different from that when a single adjustment method was used (see Tables 4 and 5).**"

6. Line 598: Rewrite the following phrase: "making it underestimates the hotspot but much closer to the VCD over the rest of the areas."
Response: Thank you. We have rewritten this sentence as follows,

"In addition, the $NO_2$ VCD from **the** simulation with **the** combined adjustments using **the** EmisAdj_sub method for emission refresh shows a similar spatial pattern **to** that  using BOE (Fig. S3). The $NO_2$ VCD level, however, is  lower over the NYC area,

**suggesting an**  underestimate over the hotspot but much  **better prediction** over the rest of the areas."

7. Lines 602-603: This may be another indication (in addition to surface ozone results) that EmisAdj_sub is indeed the better approach. Any comments?
Response: Agreed. We added the following statement to reflect this view of point:

"It indicates the advantage of adjusting emissions with a finer spatial resolution in simulating $NO_2$ vertical column in this study."

8. Line 637: "inputs play an important role" (insert "an")
Response: Inserted. Thank you very much!